# KCNQ potassium channels modulate Wnt activity in gastro-oesophageal adenocarcinomas

David Shorthouse[1], Lizhe Zhuang[2,*], Eric P Rahrmann[3,*], Cassandra Kosmidou[2], Katherine Wickham Rahrmann[3], Michael Hall[3], Benedict M Greenwood[1], Ginny Devonshire[3], Richard J Gilbertson[3], Rebecca C Fitzgerald[2], Benjamin A Hall[1]

Voltage-sensitive potassium channels play an important role in controlling membrane potential and ionic homeostasis in the gut and have been implicated in gastrointestinal (GI) cancers. Through large-scale analysis of 897 patients with gastro-oesophageal adenocarcinomas (GOAs) coupled with in vitro models, we find *KCNQ* family genes are mutated in ~30% of patients, and play therapeutically targetable roles in GOA cancer growth. *KCNQ1* and *KCNQ3* mediate the WNT pathway and MYC to increase proliferation through resultant effects on cadherin junctions. This also highlights novel roles of *KCNQ3* in non-excitable tissues. We also discover that activity of KCNQ3 sensitises cancer cells to existing potassium channel inhibitors and that inhibition of KCNQ activity reduces proliferation of GOA cancer cells. These findings reveal a novel and exploitable role of potassium channels in the advancement of human cancer, and highlight that supplemental treatments for GOAs may exist through KCNQ inhibitors.

## Introduction

The *KCNQ* (potassium voltage-gated channel subfamily Q) family of ion channels encode potassium transporters (1). KCNQ proteins typically repolarise the plasma membrane of a cell after depolarisation by allowing the export of potassium ions, and are therefore involved in wide-ranging biological functions including cardiac action potentials (2), neural excitability (3), and ionic homeostasis in the gastrointestinal tract (4). Diseases resulting from loss-of-function or gain-of-function (LoF/GoF) mutations in the *KCNQ* family are also wide-ranging, and include epilepsy (5), cardiac long and short QT syndrome (6), and autism-like disorders (7). Because of their involvement in human disease, numerous molecules that interact with them are therapeutics. KCNQ1 interacts with a family of KCNE ancillary proteins in varying tissues, but is otherwise homotetrameric (1). KCNQ2, KCNQ3, KCNQ4, and KCNQ5, however, can interact with each other and the KCNE family to theoretically form hundreds of combinations of channels, but are predominantly found in KCNQ2/KCNQ3 heteromers in the brain.

There is preliminary evidence to suggest that members of the *KCNQ* family may contribute to the cancer phenotype. KCNQ1 plays a role in colon cancer (8) and in hepatocellular carcinoma (9), and KCNQ3 is hypermutated in oesophageal adenocarcinoma (10). Furthermore, we have previously identified that *KCNQ1* and *KCNQ3* RNA expression correlates with a cancer gene expression profile (11). These all hint to the involvement of *KCNQ* genes in the pathogenesis of gastrointestinal cancers. This might be expected since membrane transport is critical to the homeostatic function of luminal epithelial cells, but so far, this has not been extensively explored outside of colorectal epithelium, where there is a reported interaction between KCNQ1 and $\beta$-catenin (8).

In this study, we investigate the mechanistic roles and therapeutic potential of the *KCNQ* family in gastro-oesophageal adenocarcinoma (GOA) by combining the study of highly annotated clinical and sequencing data sets of large numbers of patients (n = 897) with in vitro cell culture assays on relevant cell lines. We chose to study both gastric and oesophageal adenocarcinoma as they share similar aetiologies, and a current view is that they are likely to share the same origin (12, 13). We find that KCNQ activity impacts cancer cell growth through activating $\beta$-catenin and MYC via the modulation of cadherin junctions and that already clinically available drugs that interact with KCNQ channels are a promising therapeutic avenue for GOA.

## Results

### KCNQ genes are highly altered in GOAs

To fully characterise how *KCNQ/KCNE* genes are altered in GOAs, we studied all genetic alterations in a cohort of 897 patients with

---

[1]Department of Medical Physics and Biomedical Engineering, Malet Place Engineering Building, University College London, London, UK [2]Institute for Early Detection, CRUK Cambridge Centre, Cambridge, UK [3]Cancer Research UK Cambridge Institute, Li Ka Shing Centre, University of Cambridge, Cambridge, UK

Correspondence: d.shorthouse@ucl.ac.uk; b.hall@ucl.ac.uk
*Lizhe Zhuang and Eric P Rahrmann contributed equally to this work

---

adenocarcinomas of the stomach or oesophagus. We combined patient data from The Cancer Genome Atlas (TCGA) with our own oesophageal adenocarcinoma data (OCCAMS) as part of the International Cancer Genome Consortium, in which *KCNQ3* is recurrently missense mutated in 9.4% of patients (10). Cohorts were classified into oesophageal adenocarcinoma in two groups: TCGA (n = 93) (14) and a subset of our own data for which full genetic analysis has been performed (n = 378) (10), and gastric adenocarcinoma (STAD, n = 426) (15). 37% of all patients with GOAs (n = 331/897) had genetic alterations (either non-synonymous mutations or copy-number alterations) in at least one member of the *KCNQ/E* families.

From this data set, we took several orthogonal approaches to assess the role of the *KCNQ/E* family in the cancer. We calculated the genetic status of all members of the *KCNQ* and modulatory *KCNE* gene families, as well as several known driver genes in GOAs (Fig 1A). We find a large number of amplifications of *KCNQ2* and *KCNQ3* (defined as copy number > 2 times the average ploidy). Both genes are in chromosomal regions commonly amplified in GI cancers (*KCNQ2*: chromosome 20q13.3; *KCNQ3*: chromosome 8q24.22) and known to be involved in cancer progression (16, 17). *KCNQ3* in particular is located in a locus known to contain a large number of oncogenic protein-coding and lncRNA genes (Fig 1B), including *MYC*, and is significantly (adjusted *P* < 0.0001) co-associated with *MYC* amplifications (Table S1); thus, many patients amplifying *MYC* will also amplify *KCNQ3*. Overall, 112 (12%) patients have a mutation/copy-number change in *KCNQ3*, and although the 8q24 locus is a known susceptibility indicator in many cancers, *KCNQ3* has not previously been extensively explored in cancer. We also find that most alterations in *KCNQ1*, a gene already implicated as a tumour suppressor in colorectal adenocarcinomas (8), are deletions or missense/truncating mutation events, indicating that this proposed role may extend beyond the colorectal tract. We also find several, significant (adjusted *P* < 0.05), mutually exclusive alteration events within the *KCNQ* family (Table S1), notably between *KCNQ1* and *KCNQ3* (adjusted *P* = 0.007), between *KCNQ2* and *KCNQ3* (adjusted *P* < 0.001), and between *KCNQ3* and *KCNQ5* (adjusted *P* < 0.001). This pattern reveals that genetic alteration events generally occur in only a single *KCNQ* gene, so alteration to a single member may be sufficient to confer a selective advantage. Studying the patient stage, we find no observed correlation between mutations in the *KCNQ/E* family and American Joint Committee on Cancer stage where annotated (Fig S1A). At the individual cancer level (Fig S1B), cohorts have an equal ratio of mutations and copy-number changes, and no single disease (oesophageal, gastric, or colorectal adenocarcinoma) contains most of the alterations. To identify the functional significance of mutations in our cohort, we also performed dN/dS analysis (18) (Fig S1C). dN/dS ratios show that across all patients (n = 897), *KCNQ3* and *KCNQ5* appear under positive selection, that is, more commonly non-synonymously mutated than expected (dN/dS > 1, *Q* < 0.05), in OAC and STAD cohorts, respectively.

Overall, our analysis shows that *KCNQ* alterations are frequent and generally mutually exclusive, and *KCNQ2* and *KCNQ3* are located in known susceptibility loci. Missense mutations in our cohort are also under evolutionary selective pressure, and the most notable genes are *KCNQ1*, which generally is deleted and known to be a tumour suppressor in other cancers of the GI tract, and *KCNQ3*, which is under positive selective pressure in OAC, generally amplified, and on a known cancer susceptibility locus.

## Mutations in *KCNQ* genes potentially impact channel function

Having studied types of genetic alterations across GOAs, we next sought to investigate how missense mutations might alter KCNQ function. Although metrics such as dN/dS evaluate selection, this is limited to effects that can be understood from the sequence alone. It follows that if mutations are meaningful, they should be interpretable through changes in the protein structure. KCNQ channels contain six transmembrane helices (Fig S2A). Helices S1, S2, S3, and S4 make up a voltage sensor domain. The S5, pore, and S6 domains contain the gating components of the channel. To study the functional relevance of missense mutations in *KCNQ* genes, we performed statistical and computational biophysical analysis using known structural features. To increase the number of variants for statistical and structural analysis, we studied all mutations in any *KCNQ* genes from the Catalogue of Somatic Mutations in Cancer (COSMIC) (19), selecting for mutations occurring within patients from untargeted studies and with any cancer of the oesophagus, stomach, or small intestine.

We first applied statistical techniques to the 1D protein sequence to look at mutational clustering. Non-random mutational clustering (NMC) (20) applied to the location of mutations in protein sequence identified significant mutational clusters in *KCNQ1* and *KCNQ3* (Fig 2A and B); these correlate with a calculated mutational signature-based observed versus expected ratio applied along with the protein sequence. For *KCNQ1*, there is a clear hotspot of selected for mutations within the S2-S3 linker region (cluster 1.1) and within the S6 helix (cluster 1.2). *KCNQ3* contains a significant mutational hotspot within the S4 voltage sensor helix (cluster 3.1). Interestingly, mutations found in cluster 3.1 in KCNQ3 S4 (R227Q, R230C, and R236C) are known GoF gating mutations implicated in autism spectrum disorders (21, 22) (Table S2), indicating that cancers are selecting for mutations that increase KCNQ3 channel gating activity. We thus conclude that some mutations in GOA patients increase the activity of KCNQ3. Although *KCNQ1* and *KCNQ3* are the primarily clustered genes, there are additional regions of clustering in some other members of the *KCNQ* family (Fig S2B–D).

To study the structural context of the mutational clusters observed, we modelled the atomic 3D structures of KCNQ proteins. Homology models of each human member (KCNQ1–KCNQ5) were generated from the cryo-EM structure of *Xenopus laevis* KCNQ1 (Protein Data Bank ID: 5VMS) and simulated for 200 ns using atomistic molecular dynamics in a POPC membrane to validate model soundness (Fig S2E). Overlaying mutational frequency with the structures shows areas of high mutational burden, notably the S4 helix of KCNQ3 (Fig S2F). Calculation of mutational clusters in the 3D structures of each protein also reveals a statistically unlikely (*P* < 0.05) distribution of two clusters in KCNQ1 (Fig S2G), one of which is in the pore region (overlapping with cluster 1.2), and the other of which is in a known phosphatidylinositol-binding regulatory site (23), the disruption of which would reduce gating activity. As mutations in cluster 1.2 in KCNQ1 are in the vicinity of the pore, we

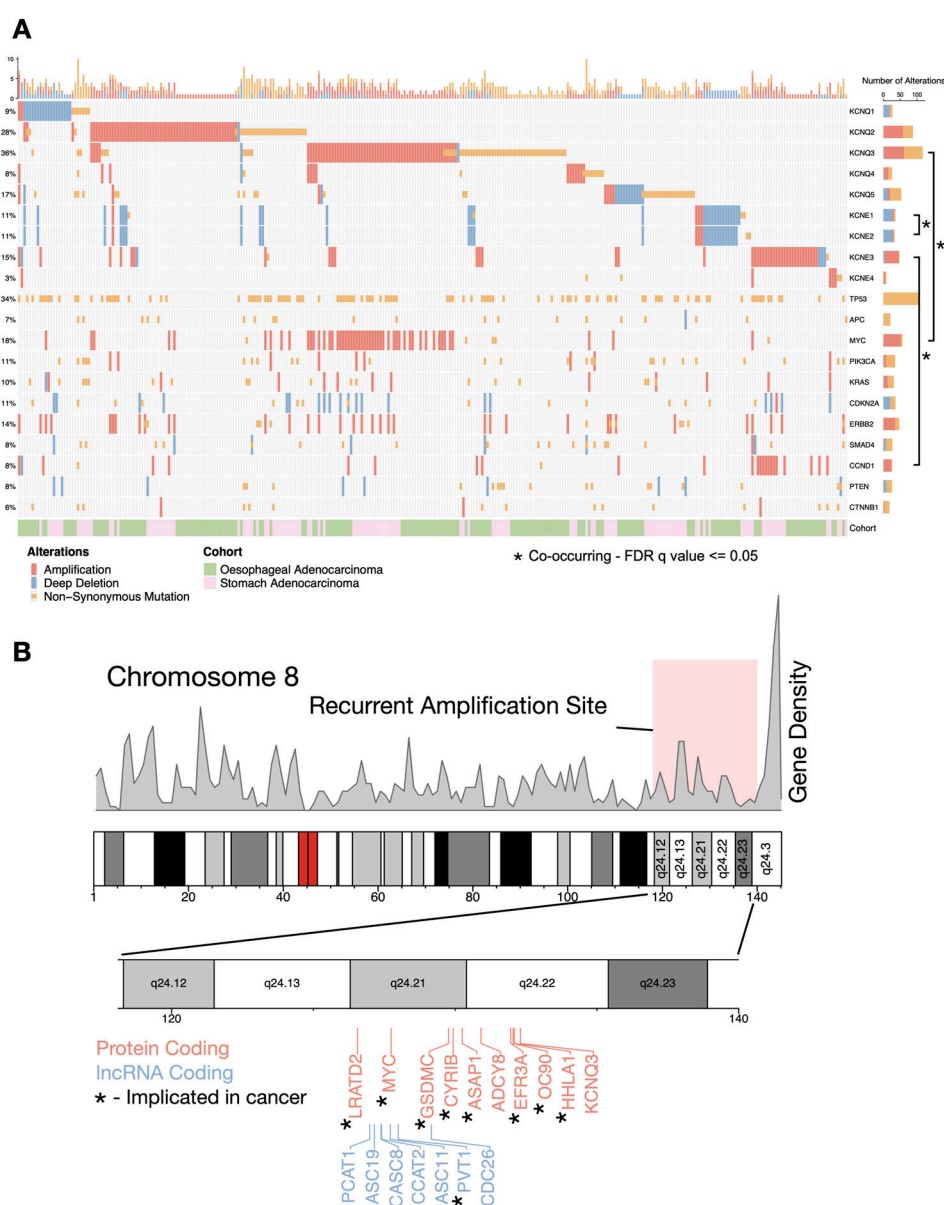

**Figure 1. *KCNQ* genes are highly altered in gastro-oesophageal adenocarcinomas.**
**(A)** Oncoprint of genetic alterations in the *KCNQ/E* gene family, and a set of known gastro-oesophageal adenocarcinoma driver genes. * represents FDR *Q*-value < 0.05 co-occurrence of alterations. **(B)** Chromosome 8q24.12-23 showing gene density, and identified genes that are recurrently amplified. * represents genes that are known drivers in human cancer.

generated models for each variant—F339L, L342F, P343L, and P343S, and an additional frequently observed mutant (A329T) (Fig 2B) within a single subunit of the channel. Pore diameter calculations show that all mutations except F339L occlude the pore, reducing or eliminating its ability to gate potassium ions, even when a single subunit is mutated, and so we conclude that mutations in cluster 1.2 are likely LOF. To assess the potential mechanism of impact of mutations to the S4 of KCNQ3, we performed molecular dynamics simulations of the single helix in a DPPC membrane using a previously developed method dubbed sidekick (24) (Fig S2H). Replication of an experimental arginine scan performed in KCNQ1 S4 demonstrates that GOF mutations tend to make the S4 more upright in the membrane and that LOF mutations result in a more tilted helical position. We find that mutations to S4 arginines consistently

change the equilibrium position of the helix similar to the GOF mutations in KCNQ1 S4.

## KCNQ channels modulate cell proliferation and correlate with clinical outcome

Based on the apparent links between *KCNQ* genomic status and cancer from patient data, we next sought to establish how changes in *KCNQ1* and *KCNQ3* expression impact cancer cell phenotype. RNA expression analysis across our cohort (n = 897) finds that *KCNQ1* is significantly down-regulated in OAC (*P* = 0.03) and slightly down-regulated in STAD (p = ns) and that *KCNQ3* is significantly up-regulated at the RNA level in both oesophageal and gastric adenocarcinomas (OAC, *P* = 0.02; STAD, *P* < 0.001) (Fig 3A)—consistent

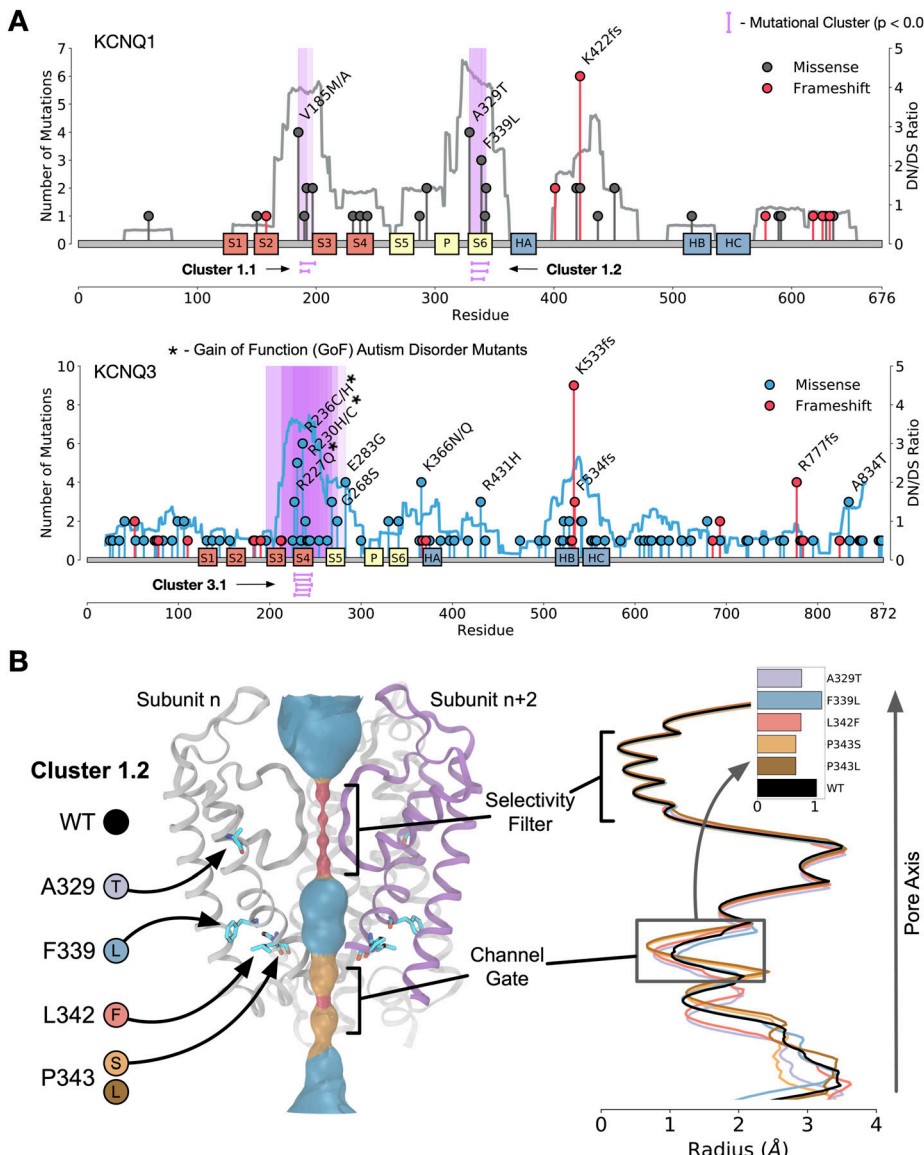

**Figure 2. Mutations in *KCNQ* genes in gastro-oesophageal adenocarcinomas alter channel function.**
**(A)** Mutational clustering for *KCNQ1* (top) and *KCNQ3* (bottom), coloured lines represent observed versus expected dN/dS ratio, and purple highlights represent statistically significant (non-random mutational clustering Q-value < 0.05) clusters of mutations. **(B)** Rendering of the pore region of KCNQ1. (Left) Mutations modelled are highlighted. (Right) HOLE analysis of the pore region of KCNQ1 WT (black) and mutations in cluster 1.2 inset is the smallest distance in the channel gate for each mutation.

with the patterns of amplification and deletion observed previously. Multivariable Cox proportional hazards analysis for the expression of *KCNQ1* and *KCNQ3*, as well as genes involved in driving GOA (Fig 3B), highlights a significant ($P < 0.01$, HR = 0.78) negative correlation between patient outcome and *KCNQ1* expression across the GOA cohort, showing *KCNQ1* expression correlates with a better prognosis. Looking at tissues, we find a positive ($P = 0.11$, HR = 1.3) correlation between *KCNQ3* expression and worse outcome in OAC, and a negative (OAC, $P = 0.12$, HR = 0.77; STAD, $P = 0.005$, HR = 0.75) correlation with the expression of *KCNQ1* (Fig S3A and B). The Kaplan–Meier analysis also reveals that patients highly expressing *KCNQ3* have a worse overall survival outcome in GOA and STAD (Fig S3C and D, GOA, logrank p = 0.09; STAD, logrank p = 0.01), and no difference in OAC (Fig S3E, logrank $P = 1.0$); across both cohorts, *KCNQ1* expression is correlated with better overall survival (Figs 3C and S3F and G, GOA, logrank p < 0.005; OAC, logrank p = 0.13; STAD, logrank p < 0.05).

Because of the co-occurrence of *KCNQ3* and *MYC* amplification, it is difficult to distinguish between effects solely caused by an increased expression of *KCNQ3* rather than an amplification of chr8q24, and so we chose to experimentally evaluate whether the expression of *KCNQ* genes can impact cancer cell phenotype in the most consistently associated cancer subtype—OAC. We chose to reduce the expression of *KCNQ1* using a CRISPR/Cas9-induced knockout (KO) (Fig S4A) and overexpression (OE) *KCNQ3* in oesophageal adenocarcinoma cell lines OE33 and FLO-1 (Fig S4B and C). KO of *KCNQ1* significantly increases the growth rate ($P = 0.003$) of OE33 cell lines (Figs 3D and S4D), but does not change growth rate in FLO1 cells. *KCNQ3* similarly significantly increases the growth rate ($P = 0.006$, though induces a small decrease in cell size—Figs 3E and S4E) when overexpressed in OE33 (Fig 3F), but induces a small decrease ($P = 0.02$) in confluence in FLO1 cells that could also correlate with cell size reductions and no change in proliferative

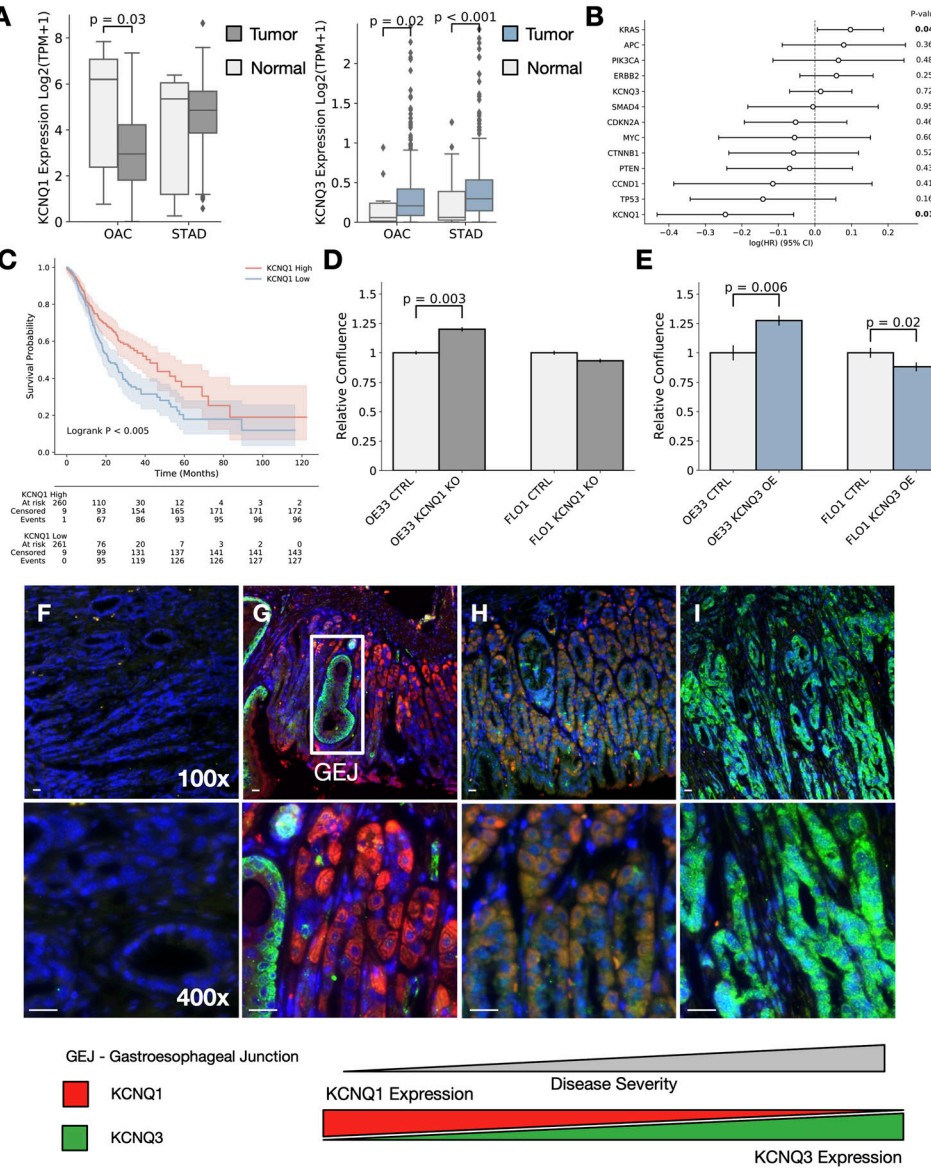

**Figure 3. *KCNQ* expression alters the gastro-oesophageal adenocarcinoma cell phenotype.**
**(A)** RNA-seq expression for *KCNQ1* and *KCNQ3* in our patient cohorts. **(B)** Multivariate Cox regression analysis of *KCNQ1* in gastro-oesophageal adenocarcinomas. **(C)** Kaplan–Meier analysis of upper and lower 50% of patients with gastric adenocarcinoma subset by KCNQ1 gene expression. **(D)** Relative confluence of cell growth in WT- versus *KCNQ3*-overexpressing (OE) OE33 and FLO1 cell lines. **(E)** Relative confluence of cell growth in WT versus KCNQ1 knockout (KO) OE33 and FLO1 cell lines. **(F, G, H, I)** Images from mouse stomach tissue. Blue represents CellTiter-Blue, red represents KCNQ1, and green represents KCNQ3. **(F, G, H, I)** Images shown are (F, G), normal Stomach; (H), benign adenoma; (I), metastatic adenocarcinoma. Scale bar represents 25 μm.

ability. This suggests *KCNQ1* expression can suppress OAC proliferation, and *KCNQ3* expression can promote it in some contexts, confirming that activity of these channels is sufficient to induce changes in cellular proliferation, prompting us to study an in vivo model, which may be more functionally relevant.

To bolster our findings and explore their generality, we looked to a murine *Prom1^{C−L}*; *Kras^{G12D}*; *Trp53^{flx/flx}* model of GOA ([25]). *Prom1* marks a stem compartment of progenitor cells that replenish tissue and cause cancers of the GI tract when mutated. Comparing the transcriptomes of isolated *Prom1+* gastric stem cells and their *Prom1−* daughter cells from normal gastric mucosa and gastric adenocarcinomas, we observe that *KCNQ1* is down-regulated and *KCNQ2/3/5* genes are significantly up-regulated (*Q* < 0.05) in gastric adenocarcinomas in this model (Fig S4F).

To validate these changes, we immunostained for KCNQ1 and KCNQ3 in *Prom1^{C−L}*; *Kras^{G12D}*; *Trp53^{flx/flx}* murine gastric mucosal tissue. Normal gastric mucosa weakly expresses KCNQ3 (green) and has the moderate expression of KCNQ1 (red) (Fig 3F and G). In benign adenoma tissue (Fig 3H), there are an up-regulation of KCNQ3 and a slight decrease in KCNQ1. In metastatic adenocarcinoma, there are an almost complete loss of KCNQ1 and a concurrent up-regulation of KCNQ3 (Fig 3I), confirming that KCNQ protein levels correlate with disease severity in a model of GOA cancer. We also find a weak but significant correlation between *KCNQ1* and *KCNQ3* expression and tumour stage in patient data, suggesting that this finding may be extended to human cancer (Fig S4G).

## KCNQ activity mediates Wnt, β-catenin, and MYC signalling

We next looked to understand how the expression of *KCNQ* genes impacts major cancer signalling pathways. We calculated the

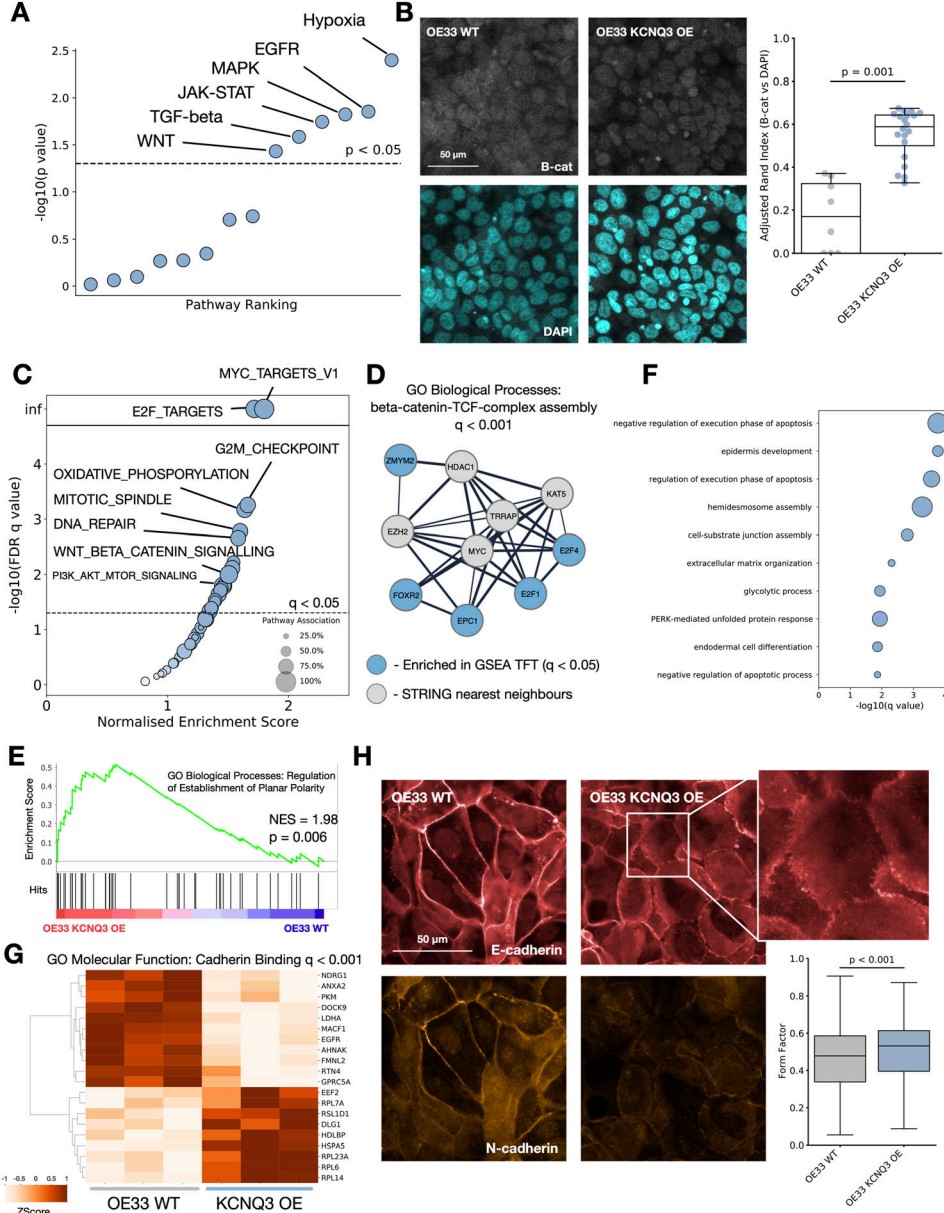

**Figure 4. *KCNQ* activity mediates *β*-catenin signalling.**
**(A)** PROGENY pathway correlation significance with *KCNQ3* RNA expression. **(B)** Imaging of *β*-catenin localisation (top—silver) and nuclear staining (bottom—blue) for WT OE33 (left) and *KCNQ3* overexpression (OE) OE33 cell lines (right). **(C)** Enrichment of hallmark gene sets by gene set enrichment analysis (GSEA) for WT versus *KCNQ3* OE OE33 cells. **(D)** String analysis of top five transcription factors identified by GSEA TFT gene sets. Genes enriched for GO biological processes identify *β*-catenin signalling (*Q* < 0.001). **(E)** GSEA enrichment plot for GO biological processes: Regulation of Establishment of Planar Cell Polarity applied to WT OE33 cell lines versus *KCNQ3* OE OE33. **(F)** GO biological process enrichment significance for significantly (*Q* < 0.05) differentially expressed genes in *KCNQ3* WT versus *KCNQ3* OE OE33. **(G)** Heatmap of genes involved in the most enriched GO molecular function (cadherin binding) for *KCNQ3* WT versus OE OE33. **(H)** Imaging of E-cadherin (red) and N-cadherin (orange) in WT OE33 (left) versus *KCNQ3* OE OE33 (right), and form factor calculation for microscopy images, N = 1,371.

PROGENY pathway scores for every patient with RNA expression data, and correlated the scores of all 14 pathways with *KCNQ1* and *KCNQ3* gene expression through a linear regression model, correcting for tissue-specific differences (Figs 4A and S5A). We find a significant correlation between *KCNQ* expression and the interlinked EGFR, MAPK, and WNT pathways, as well as an extremely strong (*P* < 0.001) link between *KCNQ3* expression and hypoxia. We also confirm an established link between KCNQ1 and *β*-catenin signalling in patients, as well as predict a similar relationship with KCNQ3, as clustering patients based on Wnt pathway genes finds a statistically significant partitioning of patients by high and low *KCNQ* expression (Fig S5B).

To validate the prediction that KCNQ3 activity may interact with the Wnt pathway, and to deconvolute *KCNQ3* expression and *MYC*

amplification in patients, we stained for the localisation of *β*-catenin in our *KCNQ3*-modulated cell lines. OE33 cells overexpressing KCNQ3 show significantly stronger nuclear localisation of *β*-catenin (median Adjusted Rand Index overlap B-cat and DAPI increase of 0.4, *P* = 0.001) when compared to WT OE33 (Fig 4B). FLO1 cells are known to already have a basal *β*-catenin activity (26), which may offer an explanation for why FLO1 *KCNQ*-modified cells do not show significant proliferation increases. To further study the effect of *KCNQ* in GOAs, we performed RNA sequencing analysis on our modulated and WT OE33 cell lines. Gene set enrichment analysis (GSEA) (27) confirms significant positive enrichment for *β*-catenin signalling in the KCNQ3 OE (Fig 4C) and *KCNQ1* KO cell lines (Table S3), as well as MYC signalling, E2F transcription factor activity, and G2M checkpoint activity—consistent with a more proliferative phenotype. Interestingly, both types of

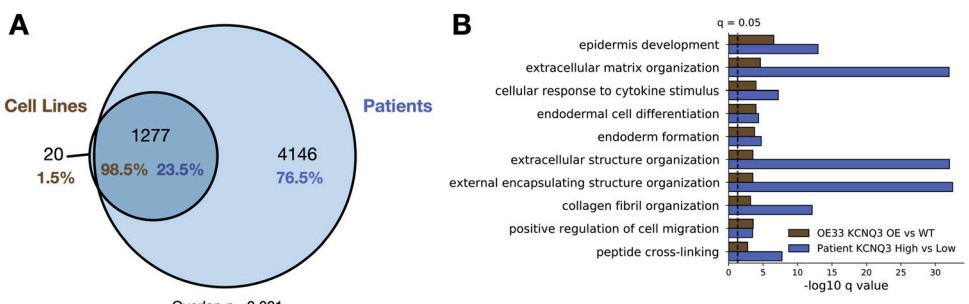

**Figure 5. Patients high and low for *KCNQ3* alter similar signalling pathways to OE of *KCNQ3* in OE33.**
**(A)** Venn diagram of overlap between enriched pathways in cell lines (*KCNQ3* WT versus *KCNQ3* OE OE33) and patients (highest 25 versus lowest 25 patients by *KCNQ3* expression in OAC). Overlap p represents using cell line pathways as custom set in g-profiler. **(B)** $-\log_{10}$ *Q*-values for the top 10 overlapping pathways between cell lines and patients.

modulated cell lines (*KCNQ1* KO and *KCNQ3* OE) show almost identical pathway alterations, suggesting that these two genes influence broadly opposite functions. Transcription factor enrichment against the TFT gene set in *KCNQ3* OE OE33 identifies a series of transcription factors linked to MYC and overlapped significantly (*Q* < 0.05) with β-catenin signalling (Figs 4D and S5C). Interestingly, as *KCNQ3* is recurrently amplified alongside *MYC*, this suggests that *KCNQ3* may act as an amplifier of *MYC* in this context, similar to the recently identified lncRNA *PVT1* (28). Finally, GSEA against the GO biological process set identifies significant enrichment for planar cell polarity pathways and non-canonical Wnt signalling (Fig 4E and Table S4)—a subtype of Wnt signalling associated with maintenance of cell polarity and known to play a role in cancer (29).

To further identify pathways altered in our cell lines, we performed differential expression analysis followed by enrichment. Differential expression confirms *KCNQ3* overexpression (Table S5), and enrichment for GO biological processes on differentially expressed (*Q* < 0.05) genes identifies biological processes including apoptosis control, cellular junctions, and cell development differentiation (Fig 4F), and clusters of differentially expressed pathways including MYC and Wnt signalling, NFKB signalling, and protein kinase C (Fig S5D). The top enriched GO molecular function in *KCNQ3* OE is cadherin binding (Figs 4G and S5E), consistent with a mechanism of action where KCNQ activity alters the structure of cadherin junctions and changes the signalling activity of β-catenin, as well as potentially activates other pathways such as NFKB or planar cell polarity.

To explore how KCNQ3 might influence planar cell polarity, we immunostained for the presence of E-cadherin and N-cadherin (CDH1 and CDH2) in our OE33 cell lines (Fig 4H) and discovered that *KCNQ3* OE results in a change in cadherin expression and cellular morphology. *KCNQ3* OE OE33 are more rounded (median form factor difference of 0.05, *P* < 0.05, N = 1,371), and many cells show the presence of membrane ruffles when E-cadherin is stained. Membrane ruffles have been observed previously and are associated with changes in cell motility and extracellular matrix organisation, and a cancer phenotype, and correlate with Wnt activity (30, 31), consistent with our RNA-seq analysis. We also find that N-cadherin expression is decreased, showing that this change is more complex than the traditional epithelial-to-mesenchymal transition.

## KCNQ channels have therapeutic potential in GOAs

Having identified that *KCNQ* expression induces cancer-associated changes in OE33 cells, we next sought to confirm these findings in

patient data. We compared GO biological process terms associated with significantly differentially expressed genes (*Q* < 0.05) for OE33 *KCNQ3* OE versus WT, and for the 25 highest and lowest *KCNQ3*-expressing patients with oesophageal adenocarcinoma (Tables S6 and S7). We find a significant (*P* < 0.0001) overlap between pathways altered in our cell lines versus patients (Fig 5A). Moreover, there is an almost complete overlap (98.5%) between pathways altered in OE33 KCNQ3 OE cell lines and patients, indicating that our cell lines accurately reproduce a subset of patient-relevant, cell-autonomous pathways. Ranking overlapping pathways by average *Q*-value (Fig 5B), the top 10 pathways include differentiation and development pathways, extracellular matrix organisation, and cell migration pathways, suggesting that patients overexpressing *KCNQ3* result in similar disruptions to cellular development and morphology as in OE33 *KCNQ3* OE cells. We find a similar trend is observed when *KCNQ1* KO versus WT pathways are compared with the top and bottom 25 OAC patients by *KCNQ1* expression (Fig S6A and B and Tables S8 and S9) (overlap between cell lines and patients of 87.1% and 65.9%, respectively, overlap *P* < 0.001), suggesting that patients with low *KCNQ1* expression alter similar pathways to *KCNQ1* KO OE33.

We surmised that modulation of KCNQ1 and KCNQ3 activity with small molecules may also confer a therapeutic benefit in GOAs. KCNQ3 represents a more feasible clinical target than KCNQ1, as it is not involved in the cardiac action potential and a number of KCNQ3-inhibiting drugs already exist. We applied two drugs to *KCNQ3* OE and WT OE33 cell lines, the KCNQ-specific inhibitor linopirdine (32), and the more broad inhibitor amitriptyline (33), which inhibits a large number of proteins including KCNQ3, is FDA-approved, and is clinically available to treat depression (34). Although linopirdine would be expected to have minor effects on KCNQ1 and KCNQ3 in these cell lines, we expect the dominant effect to be on KCNQ3, because of this protein being overexpressed in these cell lines.

Proliferation of both WT and *KCNQ3* OE OE33 was significantly reduced upon exposure to linopirdine (Figs 6A and S7A), and this effect is sensitised by the overexpression of *KCNQ3*. For FLO1 cells, however, which did not respond to *KCNQ3* overexpression, linopirdine does not have any effect—adding weight to the effect of the drug on cellular proliferation in OE33 being likely because of its actions on KCNQ3. We also find application of amitriptyline has a potent inhibitory effect on growth in OE33 cells, but that this effect is also present in FLO1. This suggests that amitriptyline likely also acts through mechanisms other than KCNQ3 to reduce the growth rate. To confirm that the linopirdine and

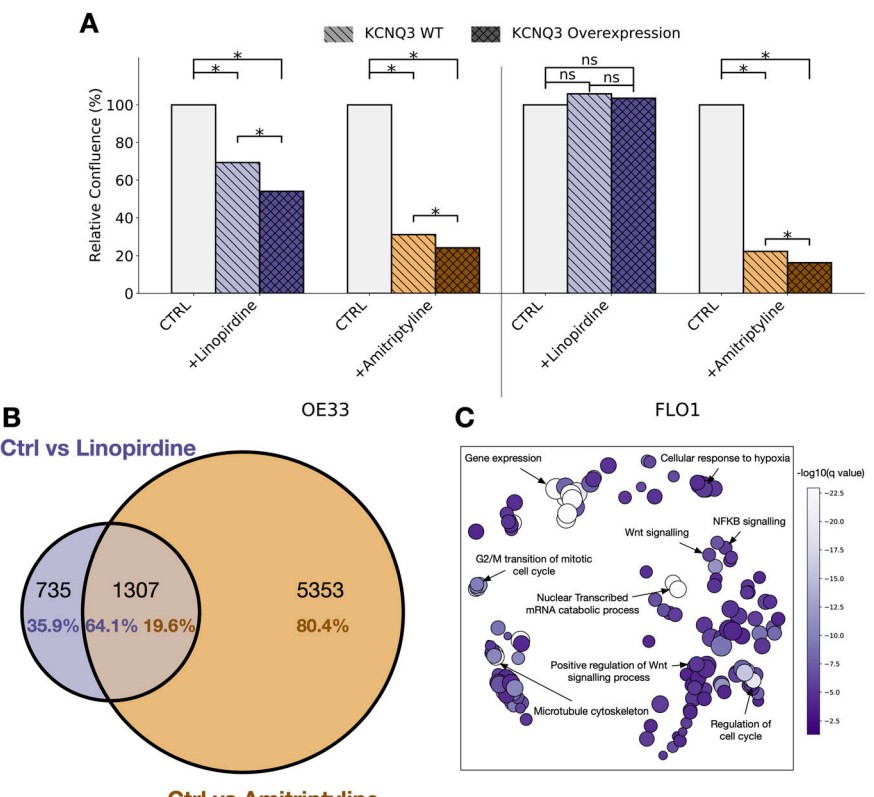

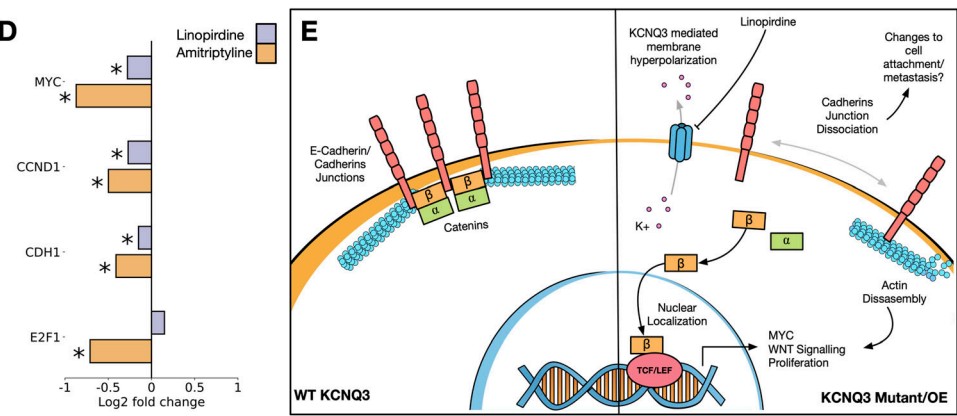

**Figure 6. KCNQ channels are potential therapeutic targets in gastro-oesophageal adenocarcinoma.**
**(A)** Relative confluence plots for OE33 (left) and FLO1 (right) cell lines exposed to linopirdine (purple) or amitriptyline (orange). Cell lines are either WT for *KCNQ3* (light) or *KCNQ3* overexpression (OE) (dark). **(B)** Venn diagram of overlapping GO biological processes enriched in ctrl versus linopirdine-exposed *KCNQ3* OE OE33 cells (purple) and ctrl versus amitriptyline-exposed *KCNQ3* OE OE33 cells. **(C)** REVIGO clustered GO biological process terms associated with overlap between ctrl versus linopirdine- and amitriptyline-exposed *KCNQ3* OE OE33 cells. **(D)** Fold change of *MYC*, *CCND1*, *CDH1*, and *E2F1* in ctrl versus linopirdine-exposed (purple) and ctrl versus amitriptyline-exposed (orange) *KCNQ3* OE OE33 cells. * represents *Q*-value < 0.05. **(E)** Speculative mechanism of *KCNQ3* activity on gastro-oesophageal adenocarcinoma cells, and their inhibition by linopirdine.

amitriptyline mechanism of action involves inhibition of KCNQ3, we also performed RNA sequencing on OE33 cells exposed to 100 mg/ml of each drug. There is a strong overlap in the differentially expressed genes associated with the application of each drug, with the KCNQ2/3-specific inhibitor linopirdine altering a subset (64.1%) of genes altered by the more broadly inhibiting amitriptyline (Fig 6B and Tables S10 and S11). Pathway enrichment and clustering with REVIGO for the overlapping gene sets (35) (Fig 6C and Table S12) identifies pathways involved in the cell cycle, WNT signalling, NFKB signalling, and the cytoskeleton as being altered in response to application of either drug, confirming that application of these inhibitors impacts cancer cell phenotype through our proposed mechanisms. We also find cadherin junctions are amongst the most enriched GO molecular functions in

both instances (Fig S7B and C). Finally, to confirm a reduction in MYC/WNT signalling in OE33 exposed to drugs, differential expression identifies a significant (*Q* < 0.05) reduction in the expression of downstream responders *MYC*, *Cyclin D1*, *E-cadherin*, and *E2F1* (Fig 6D), all of which are known players in the progression of GOAs.

# Discussion

There is emerging evidence that ion channels play a role in many and potentially all cancers (36). Therapeutics against voltage-gated potassium channels improve prognosis for glioblastoma and breast cancer (37), and studies implicate specific sodium (38) and calcium channels (39) in cancers and in specific processes such as metastasis

(40, 41). Notably, there is also a building body of work implicating potassium channels in cancer (42, 43). We show that the *KCNQ* family of genes play a significant and functional role in human GOAs. Through integration of data at the patient, cell, and protein structural levels, coupled with in vitro models, we show that *KCNQ* genes, specifically *KCNQ1* and *KCNQ3*, and their protein products contribute to gastro-oesophageal cancer phenotype and are a potential therapeutic target. We show that a large number of patients with GOAs have genetic alterations in a member of the *KCNQ* family, and expression levels of these genes are associated with patient outcome. Mutations in the *KCNQ* family have functional effects on the protein and are under selective pressure, and we find that *KCNQ* activity controls signalling activity of the WNT pathway through changing the localisation of β-catenin, and drives the cell cycle and MYC activity, as well as has a role in cell polarity as demonstrated by manipulation of these genes in cell culture. We note a large number of proteins involved in cadherin binding are differentially expressed upon the overexpression of *KCNQ3*, and future work will be necessary to unpick potential effects of KCNQ3 on cadherin junctions in GOAs; notably, one mechanism could involve cadherin junction clustering mediating β-catenin release from the inner membrane (Fig 6E). Finally, we demonstrate that *KCNQ* family members are a viable drug target with the use of already available therapeutic compounds that have not yet been actioned against cancer, but have been FDA-approved for other uses. This is particularly interesting in the case of *KCNQ3*—as it is often recurrently amplified with *MYC* and can independently drive MYC activation, it may act as a gateway to modulating the notoriously hard-to-drug MYC signalling in patients (44, 45), but we caution that correlations of *KCNQ* with patient survival may be heavily biased by MYC convolution, amidst other emerging problems identified with survival analysis (46).

By studying data from varying sources simultaneously, we find consistently that *KCNQ1* shows properties of a tumour suppressor—it is often deleted or lost in patients, mutations are generally inactivating, and cell proliferation can be increased when it is lost. Oppositingly, *KCNQ3* displays hallmarks of an oncogene, it is often amplified in cancers, mutations are mostly GoF, and cell proliferation/ Wnt signalling increases when it is overexpressed in some contexts. Thus, genes within the same family, with very similar molecular activity, have apparently opposing influences on cancer phenotype. Future work will need to be performed to unpick quite how genes within the same family appear to have opposing effects, and potential mechanisms could involve protein–protein interactions between KCNQ proteins and other membrane-associated proteins—notably other members of the *KCNQ* family. Caution must thus be taken when considering therapeutic applications of KCNQ involvement in cancer, due also to the extreme importance of *KCNQ1* in cardiac activity. Despite this, existing compounds specific to KCNQ3 (47) may have a therapeutic window, as is thought to be the case with hERG inhibitors (48). That the KCNQ2/3-specific inhibitor linopirdine shows no effect in FLO1 cell lines, but they are potently inhibited by the broader acting amitriptyline, not only indicates a key role of a number of other proteins that may be therapeutic targets in GOAs, but also opens up the possibility that patients already taking amitriptyline as an analgesic/antidepressant in cancer may be impacted by its other effects.

# Materials and Methods

## Data acquisition

TCGA level 3 data were downloaded using FireBrowse (RNA-seq) or cBioPortal (copy-number alteration, mutation, and clinical data) (49). COSMIC data (19) were downloaded from cancer.-sanger.ac.uk (version 92). We subset mutations into those only found in gastrointestinal tissue, defined as those where the primary site is in one of the following categories: "large_intestine," "small_intestine," "gastrointestinal_tract_(site_indeterminate)," "oesophagus," "stomach."

## Oncoprint

Oncoprint was generated using the oncoprint library in R (50). Copy-number alterations were determined as follows—relative copy number for each gene was defined as

$$\text{relative copy number} = log_2 \left( [\text{Total Copy Number of Gene}] / [\text{Total ploidy of sample}] \right)$$

Genes were defined as deleted if total copy number == 0 OR relative copy number < −1; genes were defined as amplified if relative copy number > 1.

## Co-association analysis

Co-association analysis was performed using DISCOVER (51).

## Chromosome plots

Chromosome plots were generated using the karyoploteR library (52).

## dN/dS

dN/dS for individual genes was calculated using the dndscv library applied to all mutations across all each tissue (OAC, STAD) (18).

For calculating the expected versus observed mutational distribution, exon data were downloaded from the Ensembl BioMart (ensembl.org/biomart). Ensembl 96, hg38.p12 was selected, and data were downloaded for chromosomes 1-22, X, and Y. We used bedtools (53) to sort data, and overlapping exons were merged. To sort data, we used the following command:

```
tail − n + 2 human_exon_bed_1 − Y.txt|cut − f 1, 2, 3|bedtools sort − i stdin > human_exon_bed_file_1 − Y_sorted.bed
```

Merging was performed using the following command:

```
bedtools merge − i human_exon_bed_file_1 − Y_sorted.bed > human_exon_bed_file_1 − Y_merged.bed
```

The R library DeconstructSigs (54) was used to generate the mutational signature for all COSMIC mutations in KCNQ genes within the selected tissues for these exons. The mutational spectrum was normalised using the mutational signature to generate the

expected relative mutation rate for each possible missense mutation. This was then multiplied by the total number of mutations in each gene to get the expected distribution of events along the gene. The observed and expected mutational frequency ratio was averaged over a sliding window of 50 bases.

### NMC

Mutational clustering was calculated using the NMC method from the R library IPAC (20) applied to the sequence alone. All mutations to each KCNQ gene in the COSMIC database for cancers of the oesophagus, stomach, and small intestine were considered. The top five mutational clusters ranked by adjusted *P*-value were plotted.

### Cox proportional hazards/KM analysis

Cox proportional hazards analysis was performed using the Python library lifelines (55). Patients were labelled with their cancer origin (OAC, STAD), and overall survival was correlated with the z-score RNA expression of *KCNQ1*, *KCNQ3*, and the previously studied driver genes (*APC*, *MYC*, *TP53*, *SMAD4*, *PIK3CA*, *KRAS*, *CDKN2A*, *CTNNB1*, *ERBB2*, *CCND1*, *PTEN*) concurrently.

The Kaplan–Meier analysis was performed using the R library Survival (56). We used the clinical data associated with TCGA and OCCAMS, which include overall survival. Survival curves were generated on the top and bottom 50% of patients ranked for the expression of each gene by z-score.

### Homology modelling/MD simulations

Homology modelling was performed using the template structure 5VMS from the Protein Data Bank (57). Sequences were aligned with MUSCLE (58) before manual adjustment based on key residues (arginines in the S4 helix, key regions of the pore domain). Single point mutations were induced in the models using the mutate_model protocol in the modeller tool as described in Feyfant et al (59), and using the mutate_model.py script available on the modeller website.

Molecular dynamics was performed using GROMACS, version 2018.1 (60).

For simulations of homology models in AT, we used the CHARMM36 forcefield (61). In each case, the protein was placed in a 15 × 15 × 15 nm box with roughly 650 DPPC lipid molecules. The set-up was performed in the same manner as systems in the MemProtMD pipeline (62). The system was converted to MARTINI coarse-grained structures (CG-MD) with an elastic network in the martiniv2 forcefield (63) and self-assembled by running a 1,000-ns molecular dynamics simulation at 323 k to allow the formation of the bilayer around the protein. The final frame of the CG-MD simulation was converted back to atomistic detail using the CG2AT method (64). The AT system was neutralised with counterions, and additional ions added up to a total NaCl concentration of 0.05 mol/litre. The system was minimised using the steepest descent algorithm until the maximum force Fmax of the system converged. Equilibration was performed using NVT followed by NPT ensembles for 100 ps each with the protein backbone restrained. We used the Verlet cut-off scheme with PME electrostatics, and treated the box as periodic in

the X, Y, and Z planes. Simulations were run for 200 ns of unrestrained molecular dynamics. Root mean square deviation was calculated for structures using the g_rmsdist command in GROMACS.

CG simulations of single helices were performed as described previously (24). Models of single helices were generated and converted to MARTINI coarse-grained structures. Helices were then inserted into DPPC bilayers and simulated for short (100 ns) simulations for 100 repeats of each sequence.

### Pore calculations

Pore analysis was performed using the algorithm HOLE (65). Pore profile was visualised using Visual Molecular Dynamics (66).

### RNA-seq processing

Quality control of raw sequencing reads was performed using FastQC v0.11.7. Reads were aligned to GRCh37 using STAR v2.6.1d. Read counts were generated within R 3.6.1 summarizeOverlaps.

### GSEA

Gene set enrichment analysis was performed using the GSEA desktop application (27). GSEA was run for 5,000 permutations, and phenotype permutations were used where the number of samples was lower than 7; otherwise, gene set permutations were performed.

### Differential expression

Differential expression analysis was performed using the R library DESeq2 (67), performed on count data. All analysis was run to compare two groups, groups were assigned within a condition matrix, and the analysis was run using the formula:

dds < −DESeqDataSetFromMatrix(countData = readcounts, colData = sample_data, design = ~Status)

### Cell culture

OE33 was cultured in RPMI (Roswell Park Memorial Institute) 1,640 medium supplied with 10% FBS, and FLO1 was cultured in DMEM supplied with 10% FBS.

### CRISPR knockout of KCNQ1

We first generated CRISPR/Cas9-expressing cell lines of OE33 and FLO1. Briefly, Lentiviral particles were generated by transfecting HEK293T cells with a Cas9-expressing plasmid, FUCas9Cherry, gift from Marco Herold (plasmid #70182 (68); Addgene), and an envelope plasmid, pMD2.G, and a packaging plasmid, psPAX2, both gifts from Didier Trono (plasmid #12259 and #12260; Addgene). OE33 and FLO1 cells were transduced with the lentivirus, subcultured, and selected for mCherry[bright] cells using FACS, respectively, to generate stable Cas9-expressing cell lines.

We then designed four sgRNA sequences targeting exon2 and exon3 of KCNQ1, which were shared by both known KCNQ1 variants, using an online tool http://crispor.tefor.net/, namely, sequences #6 CAGGGCGGCATACTGCTCGA and #7 GGCGGCATACTGCTCGATGG targeting exon2; and #8 GGCTGCCGCAGCAAGTACGT and #9 CGGCTGCCGCAG-CAAGTACG targeting exon3. sgRNA sequences were cloned into a backbone plasmid pKLV2-U6gRNA5(BbsI)-PGKpuro2ABFP-W (Fig S3A, left panel), which was a gift from Kosuke Yusa (plasmid #67974; Addgene), as described in reference 69. Briefly, for each sgRNA, two complimentary oligos were purchased, annealed, and cloned into the BbsI site of the backbone plasmid pKLV2. sgRNA lentiviruses were then generated using the aforementioned pMD2.G and psPAX2 plasmids from HEK293T cells.

OE33- and FLO1 Cas9-expressing cell lines were transduced with four sgRNA lentiviruses to generate KCNQ1 knockout cell lines, which were later subcultured and selected by puromycin treatment of 5 µg/ml for 3 d. Four KCNQ1 knockout cell lines were generated using the four sgRNA sequences. Genomic DNA was extracted from each cell line using QIAGEN AllPrep DNA/RNA Kit, and the sgRNA-targeted regions were amplified in PCR using ACCUZYME DNA Polymerase (BIO-21052; Meridian Bioscience) according to the manufacturer's manual. Primers for the PCR were as follows: TCCCCAGGTGCATCTGTGG (forward) and TCCAAGGCAGCCATGACAT (reverse) for sgRNA sequences #6 and #7 targeting exon2; and TGCAGT-GAGCGTCCCACTC (forward) and CTTCCTGGTCTGGAAACCTGG (reverse) for sgRNA sequences #8 and #9 targeting exon3. PCR products were ~200 bp long, which were run in 1% agarose gel and purified using QIAGEN Gel Extraction Kit, and then sent for Sanger sequencing provided by Source BioScience. Successful KCNQ1 knockout by the non-homologous DNA end joining was confirmed in the cell line used, sgRNA #9 (Fig S3A, right panel).

## Overexpression of KCNQ3

To generate a KCNQ3-overexpressing lentiviral plasmid, the KCNQ3 fragment was cloned from pCMV6-KCNQ3 (RC218739; OriGene) using ACCUZYME Mix 2x (BIO-25028; Meridian Bioscience) using primers of GGGCCTTCTAGAATGAAGCCTGCAGAACACGC (forward, with a XbaI cloning site) and TCACACGCTAGCTTAAATGGGCTTATTGGAAG (reverse, with a NheI cloning site).

The KCNQ3 PCR product was purified using QIAGEN QIAquick PCR Purification Kit, and then cloned into a backbone plasmid with a EGFP tag, pUltra, a gift from Malcolm Moore (plasmid #24129, (70); Addgene) between the XbaI and NheI sites (Fig S3B). Lentiviruses were then generated using the aforementioned pMD2.G and psPAX2 plasmids from HEK293T cells, which were used to transduce OE33 and FLO1 cells. Stable KCNQ3 overexpression cell lines were then generated from sorting of EGFP<sup>birght</sup> cells using FACS.

## Proliferation assay

Cells of knockout or overexpression were seeded in a 24-well plate at a density of 50,000 cells per well, four replicates per cell type or drug treatment condition. Plates were cultured in IncuCyte SX5 and scanned for the whole well using the standard phase model every 6 h. Cell confluence was quantified using the built-in Basic Analyzer and plotted over time. Each experiment was repeated at least once.

## Western blot

Cells were freshly harvested and counted. 600,000 cells were lysed using lysis buffer containing 50% of TruPAGE LDS Sample Buffer (PCG3009; Merck) and 20% 2-mercaptoethanol. Cell lysates were heated at 98°C for 5 min, cooled down to RT, diluted 1:1 using water, and run in NuPAGE precast gels (NP0321BOX; Thermo Fisher Scientific). The gels were transferred to membranes using the iBlot system (IB401001; Thermo Fisher Scientific). The membranes were incubated with primary antibodies KCNQ3 (GTX54782, 1:1,000; GeneTex) and GAPDH (ab181602, 1:10,000; Abcam) at 4°C overnight, followed by IRDye 800CW Goat anti-Rabbit IgG Secondary Antibody (925-32211, 1:5,000; LI-COR). The membranes were visualised using the LI-COR Odyssey CLx system.

## Drug treatment

Linopirdine (L134-10 MG, MW 391.46; Sigma-Aldrich) was prepared in absolute ethanol for 100 mM stock as described in reference 71. A final concentration of 50 µM was used to treat cells. Amitriptyline hydrochloride (A8404-10 G, MW 313.86; Sigma-Aldrich) was prepared in absolute ethanol for 60 mM stock as described in reference 33. A final concentration of 30 µM was used to treat cells. The culture medium was refreshed every 3 d.

## RNA-seq

Total RNA was extracted from fresh cells or mouse tissues using QIAGEN RNeasy Mini Kit. RNA-seq libraries were prepared using the Lexogen CORALL mRNA-seq kit (098.96 and 157.96) according to the manufacturer's protocol. 3 µg and 700 ng of total RNA input were used for cell line and mouse tissue RNA-seq, respectively. The libraries were sequenced in the Illumina NovaSeq platform using SR100. For cell lines, three replicates were sequenced for each cell type or drug treatment condition.

## Murine tissue immunofluorescence

Normal stomach and gastric tumour samples were harvested from *Prom1*<sup>C–L</sup>; *Kras*<sup>G12D</sup>; *Trp53*<sup>flx/flx</sup> animals (25). Tissue samples were formalin-fixed, paraffin-embedded and cut into 5-µm sections. Immunofluorescence was performed using sections of formalin-fixed, paraffin-embedded tissue generated as described above. Antigen retrieval in tissue sections was achieved using pressure cooking in citrate buffer, pH6, for 20 min. Tissue sections were incubated with primary antibodies overnight at 4°C in a humidity chamber. Primary antibodies included Kcnq1 (1:50, ab77701; Abcam) and Kcnq3 (1:50, ab16228; Abcam). After washing, tissue sections were then incubated for 1 h at RT in a secondary antibody. Secondary antibodies included Alexa Fluor 488 or 594 (1: 200, A-21206 or A-21207; Invitrogen). Dual labelling of Kcnq1 and Kcnq3 was performed as sequential stains to account for the same species with appropriate single-stain controls to monitor

for non-specific staining of each antibody. Sections were then counterstained using DAPI (1:10,000, 4083; Cell Signaling) and mounted using ProLong Gold Antifade Mountant (P36930; Thermo Fisher Scientific). Digital images of tissue sections were captured using a Zeiss ImagerM2 and Apotome microscope.

### Cell line immunofluorescence

Cells were cultured in an eight-well chamber slide (154534; Thermo Fisher Scientific) to confluence, four wells per cell type. The cells were then fixed using 4% PFA for 20 min at RT and blocked using 1% BSA. Immunofluorescent staining was performed using the primary antibody of $\beta$-catenin (ab19381, 1:100; Abcam) overnight at 4°C, and the secondary antibody of anti-mouse Alexa Fluor 647 (A-21240, 1:400; Thermo Fisher Scientific) for 1 h at RT. Nuclei were counterstained using DAPI. The cells were then imaged using a Zeiss LSM 880 confocal microscope with a 20x objective.

### Microscopy quantification

Microscopy quantification was performed using CellProfiler3 (72). For nuclear versus cytoplasmic $\beta$-catenin staining, nuclei were detected using the detect primary object command, and their overlap was measured using the measureimageoverlap tool. For quantification of the cell shape, E-cadherin was used to measure cell shape using the detect primary object command.

# Data Availability

Transcriptomic profiling is made available at the GEO repository with the accession number GSE242782. Manipulated cell lines used in this study are available on request.

# Supplementary Information

# Acknowledgements

This work was supported by the Medical Research Council (grant no. MR/S000216/2) and through a Grant-in-Aid to the MRC Cancer Unit. M Hall acknowledges support from the Harrison Watson Fund at Clare College, Cambridge. BA Hall acknowledges support from the Royal Society (grant no. UF130039). RJ Gilbertson was supported by CRUK Major Centre and Cambridge Institute Core Awards and St. Jude Children's Research Hospital/American Lebanese Syrian Associated Charities (ALSAC). EP Rahrmann was supported by Marie Skłodowska-Curie Individual Fellowship from the European Commission and Cancer Center Neurobiology and Brain Tumor Program Garwood Named Fellowship from St. Jude Children's Research Hospital. The laboratory of RC Fitzgerald was funded by a Core Programme Grant from the Medical Research Council (RG84369).

## Author Contributions

D Shorthouse: data curation, software, validation, investigation, visualisation, and writing—original draft, review, and editing.
L Zhuang: investigation and writing—original draft.
EP Rahrmann: investigation and writing—original draft, review, and editing.
C Kosmidou: investigation.
K Wickham Rahrmann: investigation.
M Hall: software and methodology.
BM Greenwood: investigation.
G Devonshire: data curation.
RJ Gilbertson: resources, supervision, and investigation.
RC Fitzgerald: conceptualisation, data curation, supervision, investigation, and writing—original draft, review, and editing.
BA Hall: conceptualisation, resources, software, supervision, funding acquisition, investigation, methodology, project administration, and writing—original draft, review, and editing.

## Conflict of Interest Statement

RC Fitzgerald is named on patents related to Cytosponge and related assays, which have been licensed by the Medical Research Council to Covidien GI Solutions (now Medtronic) and is a co-founder of CYTED Ltd. The other authors declare no competing interests.

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
