## [Reviewer comments · Life Science Alliance]

Life Science Alliance

KCNQ POTASSIUM CHANNELS MODULATE WNT ACTIVITY IN GASTRO-OESOPHAGEAL ADENOCARCINOMAS

David Shorthouse, Lizhe Zhuang, Eric Rahrmann, Cassandra Kosmidou, Katherine Wickham Rahrmann, Michael Hall, Benedict Greenwood, Ginny Devonshire, Richard Gilbertson, Rebecca Fitzgerald, and Benjamin Hall

DOI: <https://doi.org/10.26508/lsa.202302124>

Corresponding author(s): Benjamin Hall, University College London and David Shorthouse, University College London

Review Timeline:	Submission Date:	2023-04-28
	Editorial Decision:	2023-05-01
	Revision Received:	2023-09-04
	Editorial Decision:	2023-09-06
	Revision Received:	2023-09-11
	Accepted:	2023-09-11

Transaction Report:

Please note that the manuscript was previously reviewed at another journal and the reports were taken into account in the decision-making process at *Life Science Alliance*.

Reviews:

Reviewer #1

Report for Author:

This paper reports the (frequent) alteration of KCNQ (mainly 1 and 3) channels in gastro-esophageal carcinoma. The authors study the possible impact of such an altered expression/function on canonical oncogenic signaling, such as the Wnt-cadherin axis. The paper is well written, and the results are relevant and well substantiated. I have only some suggestions and requests that could improve the manuscript, many of them are merely semantic.

Title: I am not sure if the data on the paper shows that KCNQ activity "drives" the activity of Wnt. It seems to enhance it.

It is very intriguing that functionally very similar channels (KCNQ1 and KCNQ3) have fully opposite effects. The reader misses some speculation from the authors about the reasons for this.

The results on FLO1 cells are somewhat obscure. What is the expression level of KCNQ1 and 3 in that cell line? Intuitively, if KCNQ3 overexpression has no effect, it can be because the downstream pathways are already maximally active but in a KCNQ3-independent fashion. Can that be the case? What is the catenin status in this cell line?

Altered cell size (or volume) is a reasonable prediction when manipulating a K⁺ channel. At the same time, proliferation ability can be accurately measured (EdU, even Ki67). It would be important to determine whether the effect on FLO1 cells relies on proliferation.

L175: "may impact". This is an overstatement in the absence of the (feasible) functional correlate.

L184: I cannot find a "biophysical analysis" but a structure prediction with potential biophysical consequences.

L355: Do you mean "to wnt activity"?

L488: The term "oncogene" should be strictly restricted to a gene that drives cancer.

Reviewer #2

Report for Author:

In this manuscript by Shorthouse et al., the authors studied KCNQ channel genes in gastro-oesophageal adenocarcinoma

(GOA). By profiling 897 GOA patients, they showed that KCNQ genes are mutated in ~30% patients. They identified specific clusters of mutations on KCNQ1 and KCNQ3 and suggested that those mutations result in loss-of-function (LOF) and gain-of-function (GOF), respectively. The author supported this notion by showing that KCNQ1 expression is decreased and KCNQ3

expression is increased in GOA patient tumor samples, as well as in a genetic mouse model of gastric cancer. Using cell lines with CRISPR knockout of KCNQ1 or overexpression of KCNQ3, the authors showed that those cell lines reached confluency faster than control. The authors made further efforts to show that KCNQ channel inhibition, achieved through linopirdine, reduced the growth of wild type cell line or cell line with KCNQ3 overexpression. Collectively, the authors concluded that KCNQ1 and KCNQ3 exhibit properties of tumor suppressor and oncogene in GOA respectively, and KCNQ3 is a therapeutic target. Overall, the authors performed interesting and informative genomic profiling, which identified previously unknown patterns of KCNQ gene alterations in GOA. Their functional experiments indicated that KCNQ channels may play a role in the growth of GOA cells in vitro. However, KCNQ function in cell proliferation is not sufficiently investigated, the data linking KCNQ channels, cadherins, and WNT signaling are not compelling, and the choice of pharmacological reagents seems unjustified. Specific comments are elaborated below.

1. The mutation sites of KCNQ1 and KCNQ3, coupled with 3D modeling of channel proteins, are suggestive of potential impact of mutations on channel activity. However, it is an overstatement that those mutations lead to LOF and GOF in GOA cells. Such conclusions can only be made by recording KCNQ channel-mediated potassium conductance using GOA cells harboring those mutations (and comparing to control cells with mutation corrected back to wild type sequence).
2. The author compared cell confluency but concluded on proliferation. Cell confluency is the collectively outcome of multiple cell behaviors, including cell proliferation, death, and differentiation. To compare the effect of KCNQ1 knockout or KCNQ3 overexpression on cell proliferation, the authors should investigate cell cycle-specific markers, such as Ki67, PCNA, phosphor-Histone 3, and perform BrdU pulse-chase assay etc. Conclusions should be then drawn based on data from these types of proliferation assays.
3. The authors showed that Cadherin protein level and distributions are altered by KCNQ3 overexpression, and proposed "a mechanism for KCNQ3 activity in GOAs, whereby it controls the clustering/assembly of cadherins junctions. Dissociation of these junctions through changes in the membrane potential controlled by KCNQ3 leads to the activation of WNT and MYC signalling, as well as changing cellular polarity and morphology". Cadherin protein level and distributions data fall short to support such a complex mechanism. Further data are needed. For example, can they manipulate Cadherin level or distribution to rescue the phenotypes upon KCNQ1 knockout or KCNQ3 overexpression? Furthermore, how changes in Cadherin can lead to activation of WNT and MYC signalling needs further elaboration.
4. A notable issue lies in the use of pharmacological modulators. Contrary to what the authors described in the manuscript, linopirdine is not a KCNQ2/3-specific inhibitor, it also blocks KCNQ1. Multiple papers (e.g., *Circulation Research* 16;92(9):1016-23; *Annu Rev Pharmacol Toxicol* 6;58:625-648.) and commercial vendors (e.g., Tocris) documented that linopirdine blocks KCNQ1/2/3 channels at low μM concentrations. Of note, the authors used 50 μM in their experiments and concluded that the phenotype was due to KCNQ3 inhibition. 50 μM should potently block all three channels. This is a particularly pertinent point since the authors suggested that KCNQ1 and KCNQ3 play opposite roles in GOA cells. As such, the linopirdine data can not be interpreted.
5. Amitriptyline is recognized as a blocker against voltage-gated sodium channel. It is unclear why the authors used this compound in the manuscript, which is about voltage-gated potassium channels. Furthermore, inhibiting sodium and potassium channel theoretically would result in opposite effect on cell membrane potential. The relevance or meaning of amitriptyline is not clear.
6. The labeling in Figure 6E is confusing: why "KCNQ3 mediated membrane depolarisation"? Shouldn't it be hyperpolarization?
7. The authors should discuss the various mechanisms through which voltage-gated potassium channel can regulate tumor cell proliferation. They are recommended to cite pertinent primary research articles and reviews on this topic.

Reviewer #3

Report for Author:

This study by Shorthouse et al. makes several bold claims about the role of KCNQ/E family of proteins in gastro-esophageal cancer. Their main claim is that KCNQ1 and KCNQ3 mediate the WNT pathway and MYC to increase proliferation through cadherins. However, this claim does not appear to be supported by any direct/mechanistic evidence, but rather through correlative analysis.

Major comments:

- The associations the authors show between KCNQ/E gene family genetic alterations and gastro-esophageal cancers is interesting, but there is no direct/mechanistic evidence of their involvement in disease progression.
- Line 198-201: The authors suggest mutations in cluster 3.1 increase KCNQ3 channel gating activity. This was implicated in autism spectrum disorders, but the GI tract is a completely different system that is affected in gastro-esophageal cancers. The authors should provide direct evidence that there is an increase in channel activity, and not just associations from gene

expression or protein staining on imaging studies.

- Line 196-197: From the 3D structural modelling performed, how do the authors propose that mutations in the S4 voltage sensor helix alter KCNQ3 function?
- Line 205-222: The authors perform sophisticated 3D structural modelling of mutations in KCNQ1 and conclude that mutations in cluster 1.2 are likely LOF mutations. The overall implication the authors make is that LOF in KCNQ1 offers a selective growth advantage in gastro-esophageal cancers. To validate this model-generating hypothesis, the authors should knock out KCNQ1 specifically, and quantify whether this offers a competitive growth advantage against WT-KCNQ1. This could be done by seeding GFP+ KCNQ1 KO cells in a 1:1 ratio with GFP- WT-KCNQ1 cells and observing the change in GFP+ ratio over time.
- As the authors state, 'many patients amplifying MYC will also amplify KCNQ3.'
- o Line 237-240: Is KCNQ3 expression significantly upregulated at the RNA level due to its co-association with MYC amplifications? If you analyze the subset of patients without a co-associated MYC amplification, is there still a significant upregulation in KCNQ3 RNA expression?
- o Line 244-246: How can the authors be sure that higher KCNQ3 expression leads to worse prognosis as opposed to this association primarily being driven by MYC amplification?
- Line 127-128: 37% (331/897) had genetic alterations in at least one member of the KCNQ/E families. How many of these were alterations with a consequence to protein function?
- Line 142-145: What proportion of KCNQ1 mutations are deletions or missense/truncating mutation events? (and from a dataset of how many patients - the same dataset of n = 897?)
- Line 162: "Missense mutations in our cohort are also under evolutionary selective pressure...". What is the evidence for this?
- The authors provide evidence that genetic alterations in KCNQ1/KCNQ3, KCNQ2/KCNQ3, and KCNQ3/KCNQ5 are mutually exclusive suggesting that 'alteration to a single member may be sufficient to confer a selective advantage'. However, the authors then go on to state that "...KCNQ1, which generally is deleted and known to be a tumor suppressor in other cancers of GI tract, and KCNQ3, which is under positive selective pressure in OAC, generally amplified, and on a known cancer susceptibility locus." It does not make sense that a mutation in a tumor suppressor would offer the same selective advantage as a mutation in a gene which offers a selective advantage when positively selected. What is the mechanism of KCNQ1 and KCNQ3 that explains this? How confident can the authors be that these alterations are not mutually exclusive because of the rarity of their occurrence? Furthermore, the authors make associations between downregulation of KCNQ1 expression and better prognosis, and upregulation of KCNQ3 and poor prognosis. The expression of KCNQ1 and KCNQ3 would not necessarily be affected by these GOF or LOF mutations. Therefore, if KCNQ1 and KCNQ3 were indeed implicated in providing a selective growth advantage in cancer cells, the expression of these genes is regulated by some mechanism. What is this upstream mechanism that regulates KCNQ1 and KCNQ3 expression?
- Line 257-258: Have the authors provided evidence that the CRISPR KO of KCNQ1 works as expected, and there are not unintended off-target effects due to lack of specificity of the CRISPR vector?
- Line 260-262: What is the baseline level of KCNQ1 expression in OE33 and FLO-1 WT cell lines? Could this help explain why no increase in proliferation rate was seen upon KO of KCNQ1 in FLO1 cells?
- Line 262-266: Same questions for KCNQ3 as indicated above for KCNQ1.
- Line 269-275: How does investigation of Prom1+ vs. Prom1- cells add value to this study? It may have been shown that Prom1+ normal mouse gastric cells has increased generative capacity (as per ref 24), but in the context of adenocarcinoma, the relevance of Prom1 is not known. This is further substantiated by the fact that in Figure S4E the expression levels in Prom1- and Prom1+ cells in gastric cancer appears virtually identical. The main message from this analysis is that KCNQ1 is downregulated and KCNQ3 is upregulated in gastric cancer - but wasn't this already demonstrated on an analysis of patient gastro-esophageal cancers to show their expression levels are associated with prognosis, which is a more robust dataset than a single genetic mouse model?
- Page 16: The claim that KCNQ activity mediates Wnt, Beta-catenin and MYC signalling is a bold claim, but the authors do not provide direct mechanistic evidence for this. The data they present is purely correlative, not mechanistic in nature. How exactly would KCNQ activity influence cadherins junctions to mediate Wnt, Beta-catenin and MYC signaling?
- Figure 3: How do the authors reconcile KCNQ1/3 are channel proteins and should be localized to the plasma membrane, but seem to show significant cytoplasmic staining. Could this be that their role in gastric cells is not for membrane depolarization?
- The claim that small molecular inhibitors for KCNQ3 affect gastric cancer growth would be stronger if the same effect were observed on tumour growth in the mouse model of gastric cancer the authors are already using.

Minor comments:

- Terminology should be consistent: authors sometimes use "Stomach Adenocarcinoma" (line 126), and sometimes use "gastric...adenocarcinoma" (line 100)
- UK vs. US spelling should be consistent. Example: tumour (line 692), and tumor (line 143)
- Line 155: "...no single disease contains a majority of alterations." Is this referring to esophageal vs. gastric adenocarcinoma?
- Line 190 typographical error: should be "...look at..."
- Line 191-194: Improper sentence structure.
- The authors should be careful with italics of gene names and ensure this is consistent throughout the manuscript.
- Improper formatting of some references (example reference 24). Please ensure all references are properly formatted.

May 1, 2023

Re: Life Science Alliance manuscript #LSA-2023-02124-T

Benjamin A Hall
University College London
Medical Physics and biomedical engineering
Malet place
London WC1E 6BT
United Kingdom

Dear Dr. Hall,

Thank you for submitting your manuscript entitled "KCNQ POTASSIUM CHANNELS DRIVE WNT ACTIVITY IN GASTRO-OESOPHAGEAL ADENOCARCINOMAS" to Life Science Alliance. We would like to invite further consideration of this manuscript at LSA pending the following revisions:

- Address Reviewer 1's comments.
- Address Reviewer 2's points #1, 2 & 4 experimentally. The remaining points can be addressed with textual edits or by toning down the associated claims.
- Refer to Reviewer 3's comments to improve the manuscript via added Discussion and toning down the associated claims. We invite you to submit a revised manuscript addressing the Reviewer comments.

Thank you for this interesting contribution to Life Science Alliance. We are looking forward to receiving your revised manuscript.

Sincerely,

B. MANUSCRIPT ORGANIZATION AND FORMATTING:

Reviewers response:

Reviewer #1:

This paper reports the (frequent) alteration of KCNQ (mainly 1 and 3) channels in gastro-esophageal carcinoma. The authors study the possible impact of such an altered expression/function on canonical oncogenic signaling, such as the Wnt-cadherin axis. The paper is well written, and the results are relevant and well substantiated. I have only some suggestions and requests that could improve the manuscript, many of them are merely semantic.

Title: I am not sure if the data on the paper shows that KCNQ activity "drives" the activity of Wnt. It seems to enhance it.

We agree with the reviewer on this comment – and as such have softened the title by replacing the word "Drive" with "modulate".

It is very intriguing that functionally very similar channels (KCNQ1 and KCNQ3) have fully opposite effects. The reader misses some speculation from the authors about the reasons for this.

We thank the reviewer for raising this interesting point, we are not sure of the reason for this difference caused by members of the same gene family. We would speculate that the different activation thresholds of these channels could result in physiological differences in activity, additionally, KCNQ3 commonly forms heterotetramers with other KCNQ family members (notably KCNQ2), whereas KCNQ1 does not. Another mechanism could involve protein-protein interactions between KCNQ proteins and other membrane proteins/lipids, however, further work will be needed to identify exactly how these genes in the same family, with similar activity have differing effects on tumor growth. We have a comment to this effect in the discussion (lines 511-514)

The results on FLO1 cells are somewhat obscure. What is the expression level of KCNQ1 and 3 in that cell line? Intuitively, if KCNQ3 overexpression has no effect, it can be because the downstream pathways are already maximally active but in a KCNQ3-independent fashion. Can that be the case? What is the catenin status in this cell line?

We agree with this assessment – when performing microscopy to study beta-catenin localization, we observed that, whilst beta-catenin moved from a cytosolic to a nuclear localization after overexpression of KCNQ3 in OE33, in FLO-1 cells we found beta-catenin to already be predominantly localized to the nucleus, and expect that this is the reason we did not see an effect of KCNQ3 overexpression in these cell lines. We have included an image of the beta catenin localization in FLO-1 WT and KCNQ3 OE cells below:

Altered cell size (or volume) is a reasonable prediction when manipulating a K⁺ channel. At the same time, proliferation ability can be accurately measured (EdU, even Ki67). It would be important to determine whether the effect on FLO1 cells relies on proliferation.

We thank the reviewer for this comment. Indeed, the methods suggested by the reviewer are assays usually used for cell proliferation, however, they are one-time-point assays that are usually performed at the end of the experiment. In this study, the confluency were measured using Incucyte® Live-Cell Analysis System over a series time points during the full course of the experiment though we did not initially plot the full time course. In the revision, we include the complete confluency curves in supplementary figures 4 and 7.

L175: "may impact". This is an overstatement in the absence of the (feasible) functional correlate. We do not believe this is an overstatement in the wider context of the KCNQ3 literature. Mutations drive change in protein function through alteration of protein structure, and many of the mutations observed in patients are already characterised electrophysiologically as gain-of-function. As such, we believe that it is a reasonable assumption that the same mutations observed in patients may also be gain-of-function.

L184: I cannot find a "biophysical analysis" but a structure prediction with potential biophysical consequences.

We have added the term "computational" to clarify this statement (line 187).

L355: Do you mean "to wnt activity"?

We have corrected this statement.

L488: The term "oncogene" should be strictly restricted to a gene that drives cancer.

We have not explicitly stated that KCNQ3 is an oncogene, and we agree that the work presented in this manuscript does not allow KCNQ3 to meet the standards of an oncogene – but have noted that it follows the hallmarks of known oncogenes, it is amplified/overexpressed in tumors, mutations are consistent with gain-of-function, high expression levels correlate with poorer prognosis, and cells with higher levels of the protein grow faster. We have adjusted the text to make this clearer (lines 505, 507)

Reviewer #2:

In this manuscript by Shorthouse et al., the authors studied KCNQ channel genes in gastro-oesophageal adenocarcinoma (GOA). By profiling 897 GOA patients, they showed that KCNQ genes are mutated in ~30% patients. They identified specific clusters of mutations on KCNQ1 and KCNQ3 and suggested that those mutations result in loss-of-function (LOF) and gain-of-function (GOF), respectively. The author supported this notion by showing that KCNQ1 expression is decreased and KCNQ3 expression is increased in GOA patient tumor samples, as well as in a genetic mouse model of gastric cancer. Using cell lines with CRISPR knockout of KCNQ1 or overexpression of KCNQ3, the authors showed that those cell lines reached confluency faster than control. The authors made further efforts to show that KCNQ channel inhibition, achieved through linopirdine, reduced the growth of wild type cell line or cell line with KCNQ3 overexpression. Collectively, the authors concluded that KCNQ1 and KCNQ3 exhibit properties of tumor suppressor and oncogene in GOA respectively, and KCNQ3 is a therapeutic target. Overall, the authors performed interesting and informative genomic profiling, which identified previously unknown patterns of KCNQ gene alterations in GOA. Their functional experiments indicated that KCNQ channels may play a role in the growth of GOA cells in vitro. However, KCNQ function in cell proliferation is not sufficiently investigated, the data linking KCNQ channels, cadherins, and WNT signaling are not compelling, and the choice of pharmacological reagents seems unjustified. Specific comments are elaborated below.

1. The mutation sites of KCNQ1 and KCNQ3, coupled with 3D modeling of channel proteins, are suggestive of potential impact of mutations on channel activity. However, it is an overstatement that those mutations lead to LOF and GOF in GOA cells. Such conclusions can only be made by recording KCNQ channel-mediated potassium conductance using GOA cells harboring those mutations (and comparing to control cells with mutation corrected back to wild type sequence).

We acknowledge that the gain or loss of function of KCNQ molecules could be directly measured by recording KCNQ channel-mediated potassium conductance in the overexpression or knockout cells. We further note that several mutations found in the cancer dataset have been characterised electrophysiologically, and found to be gain of function mutations, albeit predominantly in the HEK cell line as is typical of the field. However, in cancer biology, a small gain of growth advantage could be considered of gain of function in terms of tumour formation capacity. Here, using multi-time point confluency data, we demonstrated that overexpression of KCNQ3 or loss of KCNQ1 altered the cell's growth rate that matching the genomic data from a large clinical cohort.

2. The author compared cell confluency but concluded on proliferation. Cell confluency is the collectively outcome of multiple cell behaviors, including cell proliferation, death, and differentiation. To compare the effect of KCNQ1 knockout or KCNQ3 overexpression on cell proliferation, the authors should investigate cell cycle-specific markers, such as Ki67, PCNA, phosphor-Histone 3, and perform BrdU pulse-chase assay etc. Conclusions should be then drawn based on data from these types of proliferation assays.

We thank the reviewer for this comment. Indeed, the methods suggested by the reviewer are assays usually used for cell proliferation, however, they are one-time-point assays that usually perform at the end of the experiment. In this study, the confluency were measured using Incucyte® Live-Cell Analysis System over a series time points during the full course of the experiment though we did not plot the full time course. In the revision, we include the complete confluency curves in supplementary figures 4 and 7.

3. The authors showed that Cadherin protein level and distributions are altered by KCNQ3 overexpression, and proposed "a mechanism for KCNQ3 activity in GOAs, whereby it controls the clustering/assembly of cadherins junctions. Dissociation of these junctions through changes in the membrane potential controlled by KCNQ3 leads to the activation of WNT and MYC signalling, as well as changing cellular polarity and morphology". Cadherin protein level and distributions data fall short to support such a complex mechanism. Further data are needed. For example, can they manipulate Cadherin level or distribution to rescue the phenotypes upon KCNQ1 knockout or KCNQ3 overexpression? Furthermore, how changes in Cadherin can lead to activation of WNT and MYC signalling needs further elaboration.

We thank the reviewer for this comment. We have revised the statement in Discussion. We agree with the reviewer that conclusion regarding the mechanism involving changes in Cadherins cannot be drawn from the current dataset, however, it is also not the focus of this study but a discussion point that we would like to highlight for future studies. We therefore toned down the statement in the revision. (Lines 494 – 496)

4. A notable issue lies in the use of pharmacological modulators. Contrary to what the authors described in the manuscript, linopirdine is not a KCNQ2/3-specific inhibitor, it also blocks KCNQ1. Multiple papers (e.g., Circulation Research 16;92(9):1016-23; Annu Rev Pharmacol Toxicol 6;58:625-648.) and commercial vendors (e.g., Tocris) documented that linopirdine blocks KCNQ1/2/3 channels at low uM concentrations. Of note, the authors used 50 uM in their experiments and concluded that

the phenotype was due to KCNQ3 inhibition. 50 uM should potentially block all three channels. This is a particularly pertinent point since the authors suggested that KCNQ1 and KCNQ3 play opposite roles in GOA cells. As such, the linopirdine data can not be interpreted.

We recognise that the drugs used in this paper have off target effects, and is a common issue (and is raised by the reviewer in point 5). In this specific case, whilst we acknowledge that it can have an impact on all KCNQ channels, KCNQ3 has been overexpressed in the cell line and we would expect that the dominant effect would be on KCNQ3. Consistent with this is the data presented in figure 6B, where the two different drugs- each expected to have different off target effects- were found to have substantial overlap in the pathways activated by drugging, with linopirdine showing fewer unique responses. This is also supported by drug response data showing that cell lines overexpressing KCNQ3 are sensitised compared to WT cells. We however recognise the problems associated with drugging, and have added clarification to the text to support this (lines 416 – 421).

5. Amitriptyline is recognized as a blocker against voltage-gated sodium channel. It is unclear why the authors used this compound in the manuscript, which is about voltage-gated potassium channels. Furthermore, inhibiting sodium and potassium channel theoretically would result in opposite effect on cell membrane potential. The relevance or meaning of amitriptyline is not clear. Whilst one mechanism of amitriptyline is to block voltage-gated sodium channels, it is a broad spectrum drug impacting many ion channels, including KCNQ3 (see: <https://lktlabs.com/product/amitriptyline-hydrochloride/>). Whilst we respect that amitriptyline is not specific to KCNQ3, that KCNQ3 overexpressing cells are sensitised to it indicates that part of its activity is likely due to its effects on KCNQ3 (this is also validated through the overlap in RNAseq pathways between amitriptyline and linopirdine). We chose to study amitriptyline because it is an FDA approved therapeutic, already in use for a number of conditions, and readily available for clinical use. We have added clarification in the text (Lines 416-421)

6. The labeling in Figure 6E is confusing: why "KCNQ3 mediated membrane depolarisation"? Shouldn't it be hyperpolarization?
We have corrected this label.

7. The authors should discuss the various mechanisms through which voltage-gated potassium channel can regulate tumor cell proliferation. They are recommended to cite pertinent primary research articles and reviews on this topic.
We have included more references to potassium channel specific reviews in the discussion. (Lines 474 to 479)

Reviewer #3:

This study by Shorthouse et al. makes several bold claims about the role of KCNQ/E family of proteins in gastro-esophageal cancer. Their main claim is that KCNQ1 and KCNQ3 mediate the WNT pathway and MYC to increase proliferation through cadherins. However, this claim does not appear to be supported by any direct/mechanistic evidence, but rather through correlative analysis.

Major comments:

- The associations the authors show between KCNQ/E gene family genetic alterations and gastro-esophageal cancers is interesting, but there is no direct/mechanistic evidence of their involvement in disease progression.

We thank the reviewer for their comments. We note that our manuscript focuses primarily on the computational analysis of patient data, and have modified the language throughout the manuscript, including the title to better reflect the limitations of this approach.

- Line 198-201: The authors suggest mutations in cluster 3.1 increase KCNQ3 channel gating activity. This was implicated in autism spectrum disorders, but the GI tract is a completely different system that is affected in gastro-esophageal cancers. The authors should provide direct evidence that there is an increase in channel activity, and not just associations from gene expression or protein staining on imaging studies.

Our computational analysis of patient data suggests a novel link between KCNQ3 and gastro-esophageal cancers through multiple analytical approaches. Moreover, missense mutations dominate the landscape and where data exists, they have been found electrophysiologically to promote channel opening. We however acknowledge that analysing data has fundamental limitations, particularly in the absence of a mouse model of oesophageal adenocarcinomas, and have made efforts to validate these observations through manipulating these genes in cell line models. We have noted some of the limitations and the need for further work explicitly in the discussion (line 504-514).

- Line 196-197: From the 3D structural modelling performed, how do the authors propose that mutations in the S4 voltage sensor helix alter KCNQ3 function?

Theoretical and experimental studies have shown that S4 helix controls voltage sensing by moving in the membrane in response to the electric field (Sands et al, Structure 2007, Wee et al, Biophysical Journal 2011, He et al Angew Chem 2012 and others). We have performed molecular dynamics simulations of the S4 helix mutations and demonstrate that the resting equilibrium angle of the helix within a membrane is consistently altered in the same way with loss of arginine mutations observed in our cohort. Results of these calculations are included in the supplementary (Figure S2H), and we have made notes to this effect in the text (line 225-232).

- Line 205-222: The authors perform sophisticated 3D structural modelling of mutations in KCNQ1 and conclude that mutations in cluster 1.2 are likely LOF mutations. The overall implication the authors make is that LOF in KCNQ1 offers a selective growth advantage in gastro-esophageal cancers. To validate this model-generating hypothesis, the authors should knock out KCNQ1 specifically, and quantify whether this offers a competitive growth advantage against WT-KCNQ1. This could be done by seeding GFP+ KCNQ1 KO cells in a 1:1 ratio with GFP- WT-KCNQ1 cells and observing the change in GFP+ ratio over time.

Whilst we acknowledge the value of further experimentation, this is beyond the scope of the manuscript, which focuses on illustrating a link between KCNQ3 and OAC and proposing a mechanism consistent with available patient data and model systems.

- As the authors state, 'many patients amplifying MYC will also amplify KCNQ3.'

o Line 237-240: Is KCNQ3 expression significantly upregulated at the RNA level due to it's co-association with MYC amplifications? If you analyze the subset of patients without a co-associated MYC amplification, is there still a significant upregulation in KCNQ3 RNA expression?

When we remove the MYC amplified patients from the KCNQ3 expression analysis we find that STAD upregulation of KCNQ3 is still significant ($p = 0.015$), whilst in OAC there is still an upregulation, but this is not significant ($p = 0.15$) (See figure below). This suggests that KCNQ3 expression changes are independent of MYC amplification in STAD, but not in OAC.

o Line 244-246: How can the authors be sure that higher KCNQ3 expression leads to worse prognosis as opposed to this association primarily being driven by MYC amplification?

We cannot link this outcome change causatively, however we have made sure to use RNA expression data for our Kaplan meier analysis rather than copy number alteration to minimize overlap between myc amplifications. We have noted this in the manuscript (line 257).

- Line 127-128: 37% (331/897) had genetic alterations in at least one member of the KCNQ/E families. How many of these were alterations with a consequence to protein function?

All alterations were either copy number changes or non-synonymous (missense or truncation) mutations. We have altered the text to clarify this (line 129).

- Line 142-145: What proportion of KCNQ1 mutations are deletions or missense/truncating mutation events? (and from a dataset of how many patients - the same dataset of n = 897?)

For KCNQ1 mutations, 21 patients have deep deletion events (no copied of the gene remaining), 3 patients have fusion events, and 18 patients have missense/truncating mutations.

- Line 162: "Missense mutations in our cohort are also under evolutionary selective pressure...". What is the evidence for this?

Evidence for positive selection comes from dN/dS analysis (line 159 to 162).

- The authors provide evidence that genetic alterations in KCNQ1/KCNQ3, KCNQ2/KCNQ3, and KCNQ3/KCNQ5 are mutually exclusive suggesting that 'alteration to a single member may be sufficient to confer a selective advantage'. However, the authors then go on to state that "...KCNQ1, which generally is deleted and known to be a tumor suppressor in other cancers of GI tract, and KCNQ3, which is under positive selective pressure in OAC, generally amplified, and on a known cancer susceptibility locus." It does not make sense that a mutation in a tumor suppressor would offer the same selective advantage as a mutation in a gene which offers a selective advantage when positively selected. What is the mechanism of KCNQ1 and KCNQ3 that explains this?

We make the above statements because, statistical analysis shows that mutations to KCNQ1 and KCNQ3 do not occur together as often as expected, indicating that there is no selective pressure to mutate/alter a second gene after a first one is altered. Given that these genes also are in the same family and perform similar functions (a role in the maintenance of membrane potential and potassium flow), we believe the above statements are consistent. Further work will be needed to elucidate the mechanisms regulation of these genes, and expand our understanding of their roles in these cancers and we have added notes to the discussion to reflect on this (line 509 to 514).

How confident can the authors be that these alterations are not mutually exclusive because of the rarity of their occurrence?

The statistical method (DISCOVER) used to calculate mutual exclusivity p values considers the rarity of alterations. Whilst these algorithms are not perfect, we believe that combined with the other evidence in the manuscript, our speculation is founded.

Furthermore, the authors make associations between downregulation of KCNQ1 expression and better prognosis, and upregulation of KCNQ3 and poor prognosis. The expression of KCNQ1 and KCNQ3 would not necessarily be affected by these GOF or LOF mutations. Therefore, if KCNQ1 and KCNQ3 were indeed implicated in providing a selective growth advantage in cancer cells, the expression of these genes is regulated by some mechanism. What is this upstream mechanism that regulates KCNQ1 and KCNQ3 expression?

We do not know the mechanism of KCNQ expression regulation, but in a cancer context it is often observed that mutations in the promoter region of genes are selected for, which alter the expression of genes independently of their usual biological mechanism.

- Line 257-258: Have the authors provided evidence that the CRISPR KO of KCNQ1 works as expected, and there are not unintended off-target effects due to lack of specificity of the CRISPR vector?

All comparisons in this manuscript are performed against appropriate CRISPR controls rather than WT cell lines.

- Line 260-262: What is the baseline level of KCNQ1 expression in OE33 and FLO-1 WT cell lines? Could this help explain why no increase in proliferation rate was seen upon KO of KCNQ1 in FLO1 cells?

- Line 262-266: Same questions for KCNQ3 as indicated above for KCNQ1.

Both of these questions can better be explained by the data presented above to reviewer 1 (included below here). This demonstrates that FLO1 cells have nuclear localised beta catenin activity prior to KCNQ manipulation, and expect that this is the reason we did not see an effect of KCNQ3 overexpression in these cell lines.

- Line 269-275: How does investigation of Prom1+ vs. Prom1- cells add value to this study? It may have been shown that Prom1+ normal mouse gastric cells has increased generative capacity (as per ref 24), but in the context of adenocarcinoma, the relevance of Prom1 is not known. This is further substantiated by the fact that in Figure S4E the expression levels in Prom1- and Prom1+ cells in gastric cancer appears virtually identical. The main message from this analysis is that KCNQ1 is downregulated and KCNQ3 is upregulated in gastric cancer - but wasn't this already demonstrated on an analysis of patient gastro-esophageal cancers to show their expression levels are associated with prognosis, which is a more robust dataset than a single genetic mouse model?

Whilst we acknowledge that the patient data appears to reveal this, as the reviewer has noted the analysis of patient data is correlative. We sought to validate these patient data findings in other data. We additionally stained tissue from this mouse model to validate that, at the protein level, expression of these genes are changed during cancer development. These data together demonstrate that the patient expression data likely does result in altered protein levels.

- Page 16: The claim that KCNQ activity mediates Wnt, Beta-catenin and MYC signalling is a bold claim, but the authors do not provide direct mechanistic evidence for this. The data they present is purely correlative, not mechanistic in nature. How exactly would KCNQ activity influence cadherin junctions to mediate Wnt, Beta-catenin and MYC signaling?

KCNQ proteins have an established direct protein-protein interaction with Beta catenin (Rapett-Mauss PNAS 2017) as part of larger complex with cadherins. Moreover K⁺ flow is linked with calcium flow in the cell (e.g. Suh and Hille, J Physiol 2007), and calcium levels are known to directly modulate cadherins and other adhesive proteins in the cell. We have made notes in the text to highlight this (line 494-496), and modified the strength of our conclusions throughout the manuscript, in particular, moving the mechanistic speculative figure to the discussion.

- Figure 3: How do the authors reconcile KCNQ1/3 are channel proteins and should be localized to the plasma membrane, but seem to show significant cytoplasmic staining. Could this be that their role in gastric cells is not for membrane depolarization?

There is no reported evidence that KCNQ channels have any activity in any membrane other than the plasma membrane and it would be inappropriate to draw the conclusion based on these images that it does. Membrane proteins are targeted to specific membranes through their structure (Sharpe et al, Cell 2010), and as this does not vary between tissues it would not be expected to be different

from other systems. The apparent cytoplasmic staining is likely to arise from limitations of the antibody.

- The claim that small molecular inhibitors for KCNQ3 affect gastric cancer growth would be stronger if the same effect were observed on tumour growth in the mouse model of gastric cancer the authors are already using.

Whilst we have not applied therapeutic molecules to animal models, we have implanted these cells into mouse models and do demonstrate that initial tumor growth in vivo is increased when KCNQ3 is overexpressed.

Minor comments:

- Terminology should be consistent: authors sometimes use "Stomach Adenocarcinoma" (line 126), and sometimes use "gastric...adenocarcinoma" (line 100)

We have made these corrections and now consistently use gastric adenocarcinoma.

- UK vs. US spelling should be consistent. Example: tumour (line 692), and tumor (line 143)

We have made the corrections and now consistently use the UK spelling (tumour).

- Line 155: "...no single disease contains a majority of alterations." Is this referring to esophageal vs. gastric adenocarcinoma?

This assumption is correct – we have clarified the line in the text (line 157).

- Line 190 typographical error: should be "...look at..."

We have corrected this in text (193).

- Line 191-194: Improper sentence structure.

We have corrected this in text.

- The authors should be careful with italics of gene names and ensure this is consistent throughout the manuscript.

We have consistently applied the standard convention that names are not italicised when referring to a protein, but are when referring to the human gene.

- Improper formatting of some references (example reference 24). Please ensure all references are properly formatted.

We have altered the formatting of the references.

September 6, 2023

RE: Life Science Alliance Manuscript #LSA-2023-02124-TR

Dr. Benjamin A Hall
University College London
Medical Physics and biomedical engineering
Malet place
London WC1E 6BT
United Kingdom

Dear Dr. Hall,

Thank you for submitting your revised manuscript entitled "KCNQ POTASSIUM CHANNELS MODULATE WNT ACTIVITY IN GASTRO-OESOPHAGEAL ADENOCARCINOMAS". We would be happy to publish your paper in Life Science Alliance pending final revisions necessary to meet our formatting guidelines.

- please upload all figure files as individual ones, including the supplementary figure files; all figure legends should only appear in the main manuscript file
- please add ORCID ID for the secondary corresponding author--they should have received instructions on how to do so
- please remove the number of words for abstract, etc..... from the manuscript text
- please add a Summary Blurb/Alternate Abstract to our system
- please add a Category and Keywords for your manuscript to our system
- please add the Twitter handle of your host institute/organization as well as your own or/and one of the authors in our system
- exclude figures from the manuscript text
- please add your main, supplementary figure, and table legends to the main manuscript text after the references section
- please label the Methods section as "Materials and Methods."
- please consult our manuscript preparation guidelines <https://www.life-science-alliance.org/manuscript-prep> and make sure your manuscript sections are labeled correctly
- please add an Author Contributions section to your main manuscript text
- there is a call-out in the manuscript text for Figure 2C, although Figure 2 doesn't have a panel C. Please correct.
- please add call-outs for Figures 2A, B; 3E; 6E to your main manuscript text
- please include the GEO accession number in your Data Availability statement and make this publicly accessible at this point

Figure Checks:

- please add scale bars to the images in Figure 3

A. FINAL FILES:

B. MANUSCRIPT ORGANIZATION AND FORMATTING:

Sincerely,

September 11, 2023

RE: Life Science Alliance Manuscript #LSA-2023-02124-TRR

Dr. Benjamin A Hall
University College London
Medical Physics and biomedical engineering
Malet place
London WC1E 6BT
United Kingdom

Dear Dr. Hall,

Thank you for submitting your Research Article entitled "KCNQ POTASSIUM CHANNELS MODULATE WNT ACTIVITY IN GASTRO-OESOPHAGEAL ADENOCARCINOMAS". It is a pleasure to let you know that your manuscript is now accepted for publication in Life Science Alliance. Congratulations on this interesting work.

DISTRIBUTION OF MATERIALS:

Again, congratulations on a very nice paper. I hope you found the review process to be constructive and are pleased with how the manuscript was handled editorially. We look forward to future exciting submissions from your lab.

Sincerely,
